# Uncertainty Quantification for LLM Function-Calling

## Abstract

Large Language Models (LLMs) are increasingly deployed to autonomously solve real-world tasks. A key ingredient for this is the *LLM Function-Calling* paradigm, a widely used approach for equipping LLMs with tool-use capabilities. However, an LLM calling functions incorrectly can have severe implications, especially when their effects are irreversible, e.g., transferring money or deleting data. Hence, it is of paramount importance to consider the LLM's confidence that a function call solves the task correctly prior to executing it. Uncertainty Quantification (UQ) methods can be used to quantify this confidence and prevent potentially incorrect function calls. In this work, we present what is, to our knowledge, the first evaluation of UQ methods for LLM Function-Calling (FC). While multi-sample UQ methods, such as Semantic Entropy, show strong performance for natural language Q&A tasks, we find that in the FC setting, it offers no clear advantage over simple single-sample UQ methods. Additionally, we find that the particularities of FC outputs can be leveraged to improve the performance of existing UQ methods in this setting. Specifically, multi-sample UQ methods benefit from clustering FC outputs based on their abstract syntax tree parsing, while single-sample UQ methods can be improved by selecting only semantically meaningful tokens when calculating logit-based uncertainty scores.

## 1 Introduction

Large Language Models (LLMs) have undergone remarkable advancements in recent years, evolving from text generators to systems capable of interacting with their environment through API calls and function execution. This capability, commonly referred to as "tool use" (Schick et al., 2023a; Qin et al., 2024; Mialon et al., 2023), enables LLMs to perform actions ranging from simple information retrieval to complex operations that modify the state of the systems they interact with such as performing monetary transactions or modifying databases. Major language models (Achiam et al., 2023; Grattafiori et al., 2024; Jiang et al., 2024; Gemini Team, 2024) now allow their users to integrate them with external tools, databases, and services through their *Function-Calling* (FC) interface (OpenAI, 2025; Google, 2025). Such interfaces have become the most widespread approach for accessing the LLM's tool-use capabilities (Carrigan, 2024). The integration of tool-use capabilities represents a leap toward autonomous AI systems that can take actions on behalf of users.

However, with this increased autonomy come increased risks and responsibilities. Unlike errors in text generation that may be less consequential, errors in tool use can result in irreversible changes, leading to data corruption, financial losses, or safety incidents (Andriushchenko et al., 2024; Aichberger et al., 2025). In this context, Uncertainty Quantification (UQ) emerges as a critical component for responsible deployment of LLM-based systems with tool-use capabilities. UQ methods aim to estimate a model's confidence in its decisions, enabling systems to abstain from actions or defer to human oversight when uncertainty is high. Effective UQ methods can serve as guardrails, preventing models from executing potentially harmful actions when they lack sufficient confidence.

Despite the clear importance of UQ in FC scenarios, there is a notable absence of systematic evaluations for assessing UQ methods in this specific context, as well as of UQ methods developed specifically for it. While substantial research has addressed uncertainty estimation for classification and question-answering tasks (Kirchhof et al., 2024; Kuhn et al., 2023; Ye et al., 2024), the tool-use setting introduces unique challenges. FC requires structured outputs that adhere to predefined syntax, often involving complex reasoning about

parameter selection and the appropriateness of specific API calls. It is therefore unclear how existing UQ methods fare in this setting.

In this paper we aim to address this gap. We conduct, to our knowledge, the first comprehensive evaluation of uncertainty quantification methods in LLM tool-use scenarios. To do so, we create a benchmark based upon the Berkeley Function Calling Leaderboard dataset (BFCL) (Yan et al., 2024), which provides a collection of FC examples, mostly in Python. Our benchmark evaluates various UQ approaches based on how well their confidence estimates predict the correctness of function calls outputted by LLMs. We reimplement several UQ methods, ranging from simple single-sample approaches based on model logits to more sophisticated multi-sample approaches that incorporate semantic equivalence between samples, and test how effectively they discriminate correct from incorrect function calls in a variety of setups.

Our findings indicate that in the FC setting, unlike in natural language Q&A settings, the more involved multi-sample methods, such as Semantic Entropy (SE), do not outperform simple single-sample UQ methods, such as the Greedy Negative Log-Likelihood (G-NLL). We then investigate if the performance of existing UQ methods in the FC setting can be increased by tailoring them to the characteristics of FC outputs. By adapting the clustering strategy in Semantic Entropy and the token selection in G-NLL, we achieve performance improvements for both multi-sample and single-sample UQ methods.

In summary, our contributions are as follows:

1. We build the first benchmark for UQ methods in Function-Calling settings.
2. We find that SE as the best multi-sample UQ method offers no clear advantage over G-NLL as the best single-sample method.
3. We analyze the properties of FC outputs and use them to devise variants of existing UQ methods that are tailored to the FC setting and improve their performance in this setting. Namely, we adapt the semantic-entailment method of SE to the FC setting by using Abstract Syntax Trees (AST) to cluster FC outputs, and adjust G-NLL and other single-sample UQ methods to the FC setting by defining semantically meaningful tokens and including only those in calculating UQ scores.

## 2 Background and Related Work

### 2.1 LLM Tool-Use and Function-Calling

Early examples of LLM tool-use, i.e., explicitly generating calls to external tools, which are then executed, were in Retrieval-Augmented Generation (Lewis et al., 2020), where the model is allowed to retrieve information from an external source (e.g., a database) before yielding a response. Since then, various LLM tool-use approaches have been explored in the literature (Yao et al., 2023; Schick et al., 2023b; Qin et al., 2023). The most widespread approach—which has been implemented in a very similar fashion into the majority of open- and closed-source LLMs (Carrigan, 2024; OpenAI, 2025; Google, 2025)—is the Function-Calling (FC) interface. In this framework, the LLM is explicitly provided with a list of functions in context. Then, the LLM can call one or more of those functions to address the user's request.

To evaluate tool-use capabilities of LLMs, many benchmarks have been proposed (Yan et al., 2024; Qin et al., 2024; Li et al., 2023; Lu et al., 2024). However, all of them focus primarily on the accuracy-oriented performance of LLMs on these tool-use-related tasks. While accuracy is important, it is an orthogonal question to UQ, which tells us which function calls the LLM is uncertain about. This, in turn, can be used to refuse to execute certain (potentially erroneous) function calls. This aspect, which is crucial from the reliability perspective, to the best of our knowledge, so far has been neglected in tool-use evaluation.

### 2.2 Uncertainty Quantification Methods for Natural Language Generation

An established way to prevent incorrect model outputs before they can cause harm is to detect them via uncertainty estimates (Chow, 1957). In problems like classification, where the output space is confined and well-structured, this can be as simple as returning the negative log likelihood (NLL) of the predicted class (Galil et al., 2023). When working with language models, there are at least two additional challenges. First,

rather than a single class label with a single NLL value, we now have sequences of tokens, each with their own NLL value. Second, multiple distinct sequences can have the same semantic meaning: the probability of generating a correct response is the probability of generating any one of a large set of semantically equivalent correct responses (Kuhn et al., 2023). In FC, such semantic equivalences might stem from, e.g., additional whitespaces or a different order of arguments. There are two main families of UQ methods that tackle these problems. Below we give an overview of both and refer to Sec. C for more details.

The first family uses only a single greedily decoded sequence (which is likely close to being the highest-probability sequence), and aggregates its individual token uncertainties, in different ways: **MAX** reports the highest negative log-likelihood (NLL) of all predicted tokens (i.e., highest per token uncertainty), **AVG** the average, and **G-NLL** the sum (Aichberger et al., 2024; Fadeeva et al., 2023; Vashurin et al., 2025). In addition to these uncertainty estimators, we also evaluate the naïve baseline **LEN** which reports the number of tokens of the greedy answer, as correctness metrics might spuriously correlate with shorter answers (Santilli et al., 2024). In case our benchmarking metrics were affected, it would be reflected in a high performance of this baseline.

The second family samples multiple sequences from the LLM to estimate the entropy of the LLM's distribution over whole sequences. A simple way of doing this is to directly estimate the Predictive Entropy (**PE**) of the distribution, based on the individual sequence's aggregated probabilities. However, this also captures uncertainty arising from sequences that are semantically identical. To account for this, Semantic Entropy (SE) (Kuhn et al., 2023; Farquhar et al., 2024) clusters samples that are different in form but have the same meaning, then estimates the entropy of the distribution of semantically different clusters. The original implementation of SE uses LLM-based entailment methods to cluster semantically-equivalent generations. However, in initial experiments we found this to lead to bad results for the highly structured FC outputs (see Sec. D.1), and furthermore it is of course computationally expensive. Thus, we use exact string matching (EXM), where two sequences get assigned the same cluster if they are exactly equal. Once the clusters are determined, we compute the entropy over the cluster distribution, weighing each cluster (1) by its relative frequency among all sequences ($\mathbf{DSE}_{EXM}$) or (2) by the cluster probabilities computed as the sum of probabilities of its member sequences ($\mathbf{SE}_{EXM}$), both following Farquhar et al. (2024).

Lastly, we also include $\mathbf{P}_{true}$ (Kadavath et al., 2022), a prompting-based method, see Sec. C.

## 3 Benchmarking UQ Performance in Function-Calling

We build upon the Berkeley Function Calling Leaderboard (BFCL) benchmark (Yan et al., 2024), which provides a collection of FC examples, mostly in Python with small parts in Java and JavaScript. The BFCL benchmark consists of multiple tasks, with a separate dataset for evaluating each one of them. Beyond being able to compare different UQ methods for FC purposes, developing a UQ benchmark based on the BFCL allows for users to simultaneously evaluate model accuracy—how often the LLM is right—and model UQ—how well the model's uncertainty predicts its correctness.

To quantify this, we evaluate how well uncertainty estimates discriminate correct from incorrect function calls in the selective prediction setting (El-Yaniv, 2010). First, we determine whether a given response is correct with Abstract Syntax Tree (AST) matching w.r.t. the ground truth response, as default in BFCL. Then, the problem is interpreted as a binary classification of whether the response is correct or not, using the UQ score calculated as the classifier-score. The UQ-performance is then quantified as the performance of this classifier. The most appropriate metric to use for this is the area under the receiver-operation characteristic (AUROC) (Fawcett, 2006). It yields a value of 0.5 for random uncertainty estimates and 1.0 for uncertainty estimates that perfectly discriminate correct from incorrect outputs. In Sec. D.3, we also include risk-coverage (rejection-accuracy) curves to illustrate what the AUROC values correspond to in terms of potential accuracy gains thanks to UQ-based abstention.

We evaluate the UQ methods of Sec. 2.2 across eight LLMs: Qwen2.5-{0.5,3,7}B-Instruct, Ministral-8B-Instruct-2410, Qwen3-4B-Instruct-2507, gemma-2-9b-it, and gemma-3-{4,12}b-it.[1] We evaluate the correctness

---

[1] We also experimented with gemma-3-1b-it, but found that it is unable to output more than one function call even when required and hence is not suited for BFCL.

of the greedy decoding response (temperature $T=0$) and draw 10 samples at $T=1$ to compute the multi-sample UQ methods. We begin in Sec. 3.1 by directly adapting the well-defined single-turn tasks from the BFCL to measure the utility of UQ methods. These tasks vary in difficulty, allowing to progressively test the LLMs' and UQ methods' performances on increasingly harder tasks. Next, in Sec. 3.2, we explain why such a straightforward analysis on each individual task, while useful, is insufficient for comprehensive UQ evaluation, and present our adaptations that allow for a more realistic evaluation. In Sec. 3.3, we evaluate how well the UQ methods work for requests that cannot be solved by the LLM given the available tools.

| Category | Task | Single-sample | | | Multi-sample | | | Other | |
| | | **MAX** | **AVG** | **G-NLL** | $\mathbf{SE}_{EXM}$ | $\mathbf{DSE}_{EXM}$ | **PE** | $\mathbf{P}_{true}$ | **LEN** |
|---|---|---|---|---|---|---|---|---|---|
| **Individual** | Simple | 0.74 ±.045 | 0.76 ±.046 | 0.77 ±.045 | 0.73 ±.049 | 0.73 ±.049 | 0.72 ±.045 | 0.69 ±.052 | 0.56 ±.052 |
| | Multiple | 0.71 ±.065 | 0.74 ±.066 | 0.76 ±.065 | 0.73 ±.073 | 0.73 ±.073 | 0.71 ±.071 | 0.67 ±.075 | 0.58 ±.081 |
| | Parallel | 0.66 ±.059 | 0.72 ±.058 | 0.71 ±.058 | 0.66 ±.059 | 0.66 ±.059 | 0.67 ±.061 | 0.62 ±.063 | 0.49 ±.064 |
| | Parallel-Multiple | 0.72 ±.048 | 0.74 ±.048 | 0.76 ±.048 | 0.73 ±.051 | 0.73 ±.051 | 0.72 ±.052 | 0.67 ±.059 | 0.55 ±.060 |
| **Combinations** | Simple + Multiple | 0.73 ±.036 | 0.75 ±.037 | 0.77 ±.037 | 0.73 ±.041 | 0.73 ±.041 | 0.72 ±.038 | 0.68 ±.042 | 0.57 ±.044 |
| | Simple + Parallel | 0.70 ±.037 | 0.71 ±.039 | 0.75 ±.036 | 0.71 ±.038 | 0.70 ±.038 | 0.67 ±.040 | 0.68 ±.038 | 0.59 ±.039 |
| | Multiple + Parallel-Multiple | 0.73 ±.037 | 0.71 ±.039 | 0.77 ±.037 | 0.74 ±.041 | 0.73 ±.041 | 0.68 ±.041 | 0.69 ±.045 | 0.64 ±.045 |
| | All Combined | 0.72 ±.026 | 0.71 ±.027 | 0.76 ±.025 | 0.72 ±.027 | 0.72 ±.027 | 0.68 ±.028 | 0.68 ±.029 | 0.61 ±.030 |
| **Unanswerable** | Simple + Irrelevance | 0.78 ±.026 | 0.80 ±.025 | 0.81 ±.025 | 0.76 ±.026 | 0.76 ±.026 | 0.79 ±.026 | 0.76 ±.026 | 0.50 ±.032 |
| | All Combined + Irrelevance | 0.74 ±.020 | 0.77 ±.020 | 0.78 ±.020 | 0.74 ±.020 | 0.74 ±.020 | 0.75 ±.021 | 0.71 ±.022 | 0.49 ±.024 |

Table 1: Results: Mean AUROC ± mean standard error (mean over eight LLMs) for individual tasks, task combinations, and unanswerable requests described in Sec. 3.1, Sec. 3.2, and Sec. 3.3 respectively. The best result in **green**, underlined second best. For per-LLM results, see Figs. 4 to 6 (Sec. D.2).

## 3.1 How well do the UQ Methods Work for Individual Tasks?

First, we evaluate how well the UQ methods work when we individually consider the tasks defined in BFCL. To this end, we use the following individual Python task datasets from BFCL in their standard form: *Simple*[2], *Multiple*, *Parallel*, and *Parallel-Multiple*. *Simple* consists of 400 requests where only one function is made available to the LLM. It tests whether the model is able to call the function provided, together with the correct argument values, to solve the request. *Multiple* provides definitions of multiple functions and tests whether the model can choose the correct function as well as their correct arguments to solve the request (200 requests). *Parallel* requires the model to call multiple functions in parallel, including their correct arguments (200 requests). Like in *Simple*, the model is only provided with the functions it needs to use. *Parallel-Multiple* combines the challenges of *Parallel* and *Multiple*: the model is provided with multiple functions, some of which are unnecessary to solve the request. The model needs to choose which functions to call, and set their parameters correctly (200 requests). Importantly, all of the requests in these four tasks are solvable given the functions provided as part of the request.

Throughout our analysis, we exclude the requests for which the greedy decoding results in an output that could not be interpreted by the Python interpreter, i.e., that result in an Abstract Syntax Tree (AST) decoding error. These cases are clearly incorrect since they do not produce valid function calls, but also since they do not result in an execution, they cannot cause irreversible undesired outcomes. In a practical context, we would thus not need model uncertainty to flag potentially erroneous function calls for these cases. These errors are rare (on average 3.4% of outputs on the *All Combined* split), and including them does not change the relative ranking of UQ methods (Spearman correlation 0.94); see Sec. F for effective sample sizes per model and further details.

In Table 1 (top), we report the mean (over eight LLMs) AUROC for each combination of task and UQ method, along with the mean of the per-LLM standard error. Standard errors imply a confidence interval on the mean AUROC value, and were estimated via bootstrap sampling with 1000 samples. Fig. 4 of Sec. D.2 breaks down the results in Table 1 to show the per-model performances.

---

[2]BFCL's Simple split consists of 3 parts: Python, Java, and JavaScript. To be able to compare with the other three Python-based tasks (which do not have Java & Javascript counterparts), we focus on the Python part of Simple.

The results show that the sequence likelihood (G-NLL) generally performs best (often by a small margin) and outperforms multi-sample methods. Overall, the level of performance achieved by the best methods on these individual tasks (AUROC of $\sim 0.7 - 0.8$) resembles the performance achieved by these methods on natural language Q&A tasks (Aichberger et al., 2024; Farquhar et al., 2024).

## 3.2 How well do UQ Methods Work on more Realistic Multi-Modal Task Distributions?

In accuracy-oriented evaluation, it is possible to evaluate a model on individual tasks and then obtain a valid estimator of the performance on the mixture distribution by taking a weighted average of individual task performance estimates (with weights set to the components' weights of the mixture), because accuracy is a linear metric.

However, unlike accuracy, AUROC measures ranking performance rather than raw correctness counts, and so is not a linear metric. This implies that if we consider a mixture of multiple different task distributions, we might obtain a lower AUROC performance than for any individual task alone. This is important when using a confidence threshold to decide whether to execute a function call; in general, it would not be practical to set a different threshold for different modes of the task distribution. For this reason, unlike in the accuracy-oriented benchmarks like BFCL, in order to estimate the UQ performance in a way that informs practical applications, we need to evaluate the UQ methods on more complex, multi-modal data distributions. To this end, we combine the individual BFCL task data sets described in Sec. 3.1, and evaluate the UQ methods on these combinations.

The four BFCL tasks we consider offer an opportunity to gradually increase the multi-modality of the input- and output- distributions along two axes: A) number of functions at the model's disposal; B) number of function calls the model has to produce in order to solve the task. *Simple+Multiple* vary A, while keeping B fixed at 1. *Simple+Parallel* vary B between 1 and 2, while A is always set to the same value as B. *Multiple+Parallel-Multiple* is analogous to *Simple+Parallel* in B, but with A set to a larger number such that there are always some unnecessary functions present in the input. *All Combined* combines both axes of variation.

We present the results in Table 1 (middle) and Fig. 5. At a high level, the results are similar to the individual task results. G-NLL performs best out of all of the considered methods, and on average, performance levels remain the same.

## 3.3 Do UQ Methods Allow for Identifying Requests that are Impossible to Answer?

Beyond the four task datasets introduced in Sec. 3.1, the BFCL benchmark also includes an *Irrelevance* task to test whether models can recognize and explicitly communicate when they are not able to solve the task given the provided functions. Each request in this task includes a single function definition, which is not sufficient to solve the request. Thus, a successful answer by the model for this task consists of refusing to output any function call. Note that a model is able to get perfect accuracy on this task by always refusing (independent of the request). Hence, evaluating on this task alone can yield misleading results about UQ methods. We thus combine *Irrelevance* with the other tasks, once with the *Simple* data only (*Simple + Irrelevance*), and once with all other datasets combined (*All Combined + Irrelevance*).

We present the results in Table 1 (bottom) and Fig. 6. Again, we observe good and broadly comparable performance of most UQ methods. We also find that the single-sample methods (G-NLL, AVG) slightly outperform the multi-sample methods ($SE_{EXM}$, $DSE_{EXM}$, PE).

## 3.4 Are UQ Methods Calibrated for FC?

Beyond being able to distinguish incorrect from correct outputs, which we have measured with AUROC in the previous sections, a desirable quality of a UQ score is to be calibrated, i.e., to correctly predict the probability of a sample being correct, which is important in decision-making settings (Kiyani et al., 2025). Among several calibration measures proposed, we choose smoothECE (Błasiok & Nakkiran, 2023), which is a smoothed version of expected calibration error (ECE) that does not exhibit discontinuous behavior.

Since not all of the UQ methods we evaluate yield a valid probability score, we can only evaluate calibration for the following subset: MAX, AVG, G-NLL, and $P_{true}$. In Fig. 1, we show the results for the *All Combined* task as representative for all tasks. While quantitative results differ per task, qualitatively, they stay largely the same across tasks, so we relegate the detailed plots to Sec. D.4.

We make several observations. Firstly, AVG achieves the worst calibration among all methods, being largely overconfident with confidence values (x-axis) being higher than the accuracy (y-axis). $P_{true}$ and G-NLL, on the other hand, are underconfident, as indicated by their confidence being mostly lower than the accuracy. For G-NLL, this would be expected from natural language, where the multiplication of per-token probabilities, each $< 1$, tends to lead to small sequence probabilities. However, in function calling, this phenomenon is not as extreme since most

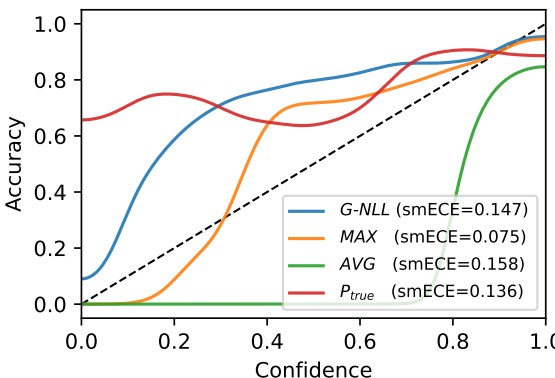

Figure 1: SmoothECE for *All Combined* task, aggregated over all models.

probabilities are $\approx 1.0$ (Sec. 4.2). Lastly, MAX achieves the best calibration. It is still slightly underconfident for most parts, but since by definition $\max_i \log p_i \geq \sum_i \log p_i$, it is less underconfident than G-NLL, especially in low-confidence regions. These insights complement the previous AUROC results, and illustrate that ECE does not necessarily correlate with AUROC.[3] In the setting of selective prediction—generating a function call with an option to abstain—the crucial property to assess is the ability to discriminate between correct and incorrect outputs, which is measured by AUROC. ECE measures whether confidence values are calibrated as probabilities, which is a secondary consideration in this context. Furthermore, all multi-sample entropy-based methods do not output probabilities, and hence their calibration cannot be evaluated. We note that for methods with high AUROC but poor calibration, post-hoc calibration techniques (e.g., Platt scaling) can be applied to improve calibration without affecting discriminative performance.

# 4 Adapting UQ Methods to FC

The results from Sec. 3 suggest that more involved multi-sample UQ methods underperform compared to simple single-sample methods. This contrasts with observations in natural language generation tasks. In the following, we investigate two hypotheses around this observation. In this section, we investigate whether the performance of existing UQ methods in the FC setting can be improved by adapting them to the particular structure of FC outputs.

## 4.1 Adapting Multi-Sample UQ Methods

In the following, we investigate whether multi-sample UQ methods can be tailored to the FC setting. Specifically, we focus on the two variants of Semantic Entropy (SE and DSE) as the highest-performing multi-sequence methods. We hypothesize that the clustering based on exact string matching (EXM) used in our experiments might be overly strict, as it does not take permutation invariance in the function arguments into account. To circumvent this, we utilize the Abstract Syntax Trees (**AST**) to cluster FC outputs based on Yan et al. (2024). Two outputs get assigned the same cluster if their AST matches. This approach effectively treats the arguments to the function as a dictionary that is permutation-invariant.

**Results**   Table 2 suggest that AST clustering improves performance over simple EXM clustering in eight out of ten tasks. However, these gains remain insufficient to establish an advantage over single-sample methods. We note that multi-sample methods also require proportionally higher throughput (proportional to the number of samples $J$), which is non-negligible especially in on-device scenarios. We further ablate the choice of temperature $T$ and number of samples $J$ in Sec. H: while increasing $T$ to 1.5 or $J$ to 30 can close

---

[3]For another illustrative example, consider the extreme case of a model with accuracy 0.5 for which a UQ method assigns a correctness probability of 0.5 for all samples. It has a perfect ECE of 0 but an AUROC of 0.5.

the gap to G-NLL, it does not surpass it on most tasks. This raises the question of why multi-sample UQ methods do not perform better in comparison, despite incorporating multiple samples. To better understand this limitation, we conduct a more detailed analysis in the following.

| Task | $\mathbf{SE}_{EXM}$ | $\mathbf{SE}_{AST}$ | $\mathbf{DSE}_{EXM}$ | $\mathbf{DSE}_{AST}$ |
|---|---|---|---|---|
| Simple | 0.73 $_{\pm.049}$ | 0.75 $_{\pm.047}$ | 0.73 $_{\pm.049}$ | 0.75 $_{\pm.047}$ |
| Multiple | 0.73 $_{\pm.073}$ | 0.75 $_{\pm.067}$ | 0.73 $_{\pm.073}$ | 0.74 $_{\pm.067}$ |
| Parallel | 0.66 $_{\pm.059}$ | 0.69 $_{\pm.056}$ | 0.66 $_{\pm.059}$ | 0.69 $_{\pm.056}$ |
| Parallel-Multiple | 0.73 $_{\pm.051}$ | 0.74 $_{\pm.049}$ | 0.73 $_{\pm.051}$ | 0.73 $_{\pm.049}$ |
| Simple + Multiple | 0.73 $_{\pm.041}$ | 0.75 $_{\pm.038}$ | 0.73 $_{\pm.041}$ | 0.75 $_{\pm.038}$ |
| Simple + Parallel | 0.71 $_{\pm.038}$ | 0.73 $_{\pm.036}$ | 0.70 $_{\pm.038}$ | 0.73 $_{\pm.036}$ |
| Multiple + Parallel-Multiple | 0.74 $_{\pm.041}$ | 0.76 $_{\pm.038}$ | 0.73 $_{\pm.041}$ | 0.75 $_{\pm.038}$ |
| All Combined | 0.72 $_{\pm.027}$ | 0.74 $_{\pm.026}$ | 0.72 $_{\pm.027}$ | 0.74 $_{\pm.026}$ |
| Simple + Irrelevance | 0.76 $_{\pm.026}$ | 0.75 $_{\pm.025}$ | 0.76 $_{\pm.026}$ | 0.74 $_{\pm.025}$ |
| All Combined + Irrelevance | 0.74 $_{\pm.020}$ | 0.74 $_{\pm.020}$ | 0.74 $_{\pm.020}$ | 0.73 $_{\pm.020}$ |

Table 2: Mean AUROC, $\pm$ mean standard error (mean taken over the eight LLMs) for all tasks from Sec. 3.

## 4.2 Patterns of Token Probabilities in FC vs. Natural Language Tasks

To better understand why SE and DSE still do not outperform single-sample methods like G-NLL, we investigate an example of a wrong response by Qwen2.5-7B-Instruct to a request from the *Parallel-Multiple* task in Fig. 2 (full description of question and available functions in Sec. A.4). Here, the LLM makes the mistake of adding an unexpected parameter at the end of the last function call (`year=1882`). This example illustrates two patterns that we find to hold in general:

1. The LLM assigns high probabilities (probability $\approx 1$) to most tokens and exhibits only a minimal degree of uncertainty at a small number of tokens ($< 5$), including those associated with the error.
2. Even for tokens the LLM is slightly uncertain about (probability $< 1$), the second most likely token is either semantically equivalent (`=["` vs. `=['`) or is significantly less likely (`year` vs. `specific`).

As a result, the samples for computing the multi-sample methods are all semantically identical to the greedy-decoded one, differing only in syntactically equivalent tokens `=["` vs. `=['`. Consequently, only a single semantic cluster is formed, resulting in low SE, even though the response is incorrect.

We compare this to a wrong response the same LLM gives to a natural language question from Natural Questions (NQ) dataset Kwiatkowski et al. (2019) in Sec. B. In this setting, neither of the above two patterns holds. In the FC setting, where available functions are provided in-context and the valid output space is restricted, the probabilities of individual tokens are higher than in natural language settings. Indeed, across the eight LLMs, we find that the average token probability on *All Combined* is 0.97,[4] while for the NQ dataset it is only 0.71. Similarly, the mean number of semantic clusters formed and corresponding SE estimates are significantly lower on BFCL than on NQ (3.4 vs. 6.4).

In contrast, single-sample methods like G-NLL rely directly on the sequence probability, making them sensitive to variations in the few tokens where the LLM assigns probabilities lower than 1. However, Fig. 2 highlights a limitation of single-sample methods in their current form: The impact of lower probabilities for semantically unimportant tokens (the token `=["` is required by the function description) might distort their score. Our knowledge about the structure of function calls and which tokens carry semantic weight raises the question of whether single-sample methods can be adapted to better leverage these characteristics of FC outputs.

## 4.3 Adapting Single-Sample UQ Methods

In the following, we implement a method that aims to address the limitation of single-sample methods. We first analyze which tokens are semantically meaningful in the FC context (as opposed to syntactically required ones) and derive a general algorithm to extract those from an FC output.

---

[4]This also explains the large overconfidence of AVG in Sec. 3.4, as the average accuracy on *All Combined* is $< 0.9$.

**Semantically Meaningful Tokens**  We define Semantically Meaningful Tokens in the FC context as those corresponding to positions where the LLM has to decide between tokens that likely lead to either correct or incorrect outputs.  Using the example response in Fig. 2, these are:

1. ■ Tokens deciding if functions are called or the task is refused (the first token: `[`)

2. ■ Tokens deciding if the right function is called (`history`, `get`, `get`)

3. ■ Tokens deciding if the right parameter is used ( `country`, `start`, `end`, `event`, `(s`, `artist`, `(s`, `artist`, `year`).

4. ■ Tokens deciding if the right parameter value is chosen ( `1`, `8`, `0`, `0`, `1`, `9`, `0`, `0`, `War`, `,`, `E`, `conomy`, `The Thinker`, `August`, `e`, `Rod`, `in`, `The`, `Kiss`, `August`, `e`, `Rod`, `in`)

5. ■ Tokens deciding if the right number of functions is called (`])`,  `,)`,  `)]`)

6. ■ Tokens deciding if the right number of parameters are included in a call (`",`,  `,`,  `,`,  `])`, `",`, `",)`

All remaining tokens are primarily syntactically required to form a correct function call output (such as `=`), or continuations of function names according to the available names in the context (such as `cul`, `pt`, `ure`). Under this definition, in the above example, the semantically meaningful tokens would include the token `year`, which has higher uncertainty and is related to the LLM's mistake. In contrast, the token `=["` would not be included, which should not have high uncertainty, since the function format requires the parameter value in the form of an array, making this token syntactically necessary. While this logic can be argued about and other ways of defining semantically meaningful tokens could be considered, we find that even this simple heuristic already yields improvements in UQ performance. Lastly, we highlight that this approach is related to previous work that proposes to focus on relevant tokens when computing UQ scores Duan et al. (2023); Fadeeva et al. (2024); Bakman et al. (2024). However, these methods are tailored to the context of natural language and no such approach has been proposed for the FC setting so far.

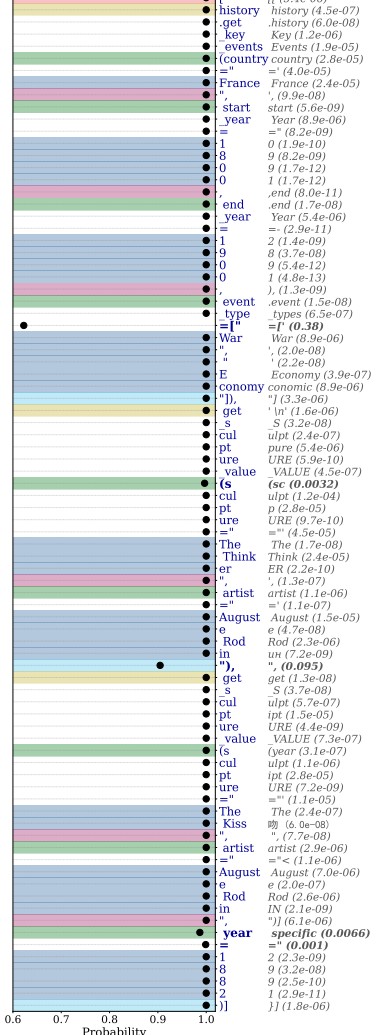

Figure 2:  Top token probabilities for an incorrect Qwen2.5-7B-Instruct response (lowest 5 in bold) in the *Multiple-Parallel* task (details in Sec. A.4). Second-highest tokens shown in grey. Color coding explained in Sec. 4.3. The Ground truth answer does not contain the argument `year=1882`.

**Implementing SMT Extraction**  Once defined, there are many options to extract the semantically meaningful tokens from a given LLM function-call output. We opt for a simple, rule-based approach that is both computationally efficient and easy to implement. Nonetheless, it requires a separate implementation for different FC output formats, depending on the LLM's specific instruction used for finetuning. For the models evaluated, gemma and Qwen models share the same FC output format corresponding to a list of Python calls (see Fig. 2), while Ministral outputs its function calls in a JSON format and thus requires a separate implementation of the token extraction function. We include the algorithms for both in Sec. G.

We note that the extraction logic follows from the formal grammar of the output schema/format used for AST parsing, and only a handful of FC output schemas account for the majority of LLM FC use. To further

demonstrate the ease of adapting the SMT extraction to new formats, we prompted GPT-5.2 with 10 examples each of Qwen7B and Ministral function-call outputs together with our Python-format SMT algorithm, asking it to produce the corresponding algorithm for the Ministral JSON format. The resulting function selected on average 57% of Ministral-generated tokens as semantically meaningful, compared to 54% by our manually written algorithm, selecting nearly identical tokens. The performance impact was minimal: on average 0.001 AUROC difference over the 10 splits, with a maximum divergence of 0.004 (on *Simple*).

Once the semantically meaningful tokens are defined and extracted, we aggregate the corresponding log-probabilities in the same way as for the whole token sequence (MAX, G-NLL, AVG). This leads to an adaptation of single-sample methods to the FC setting we refer to as *Semantically Meaningful Tokens* (**SMT**).

**Results** Table 3 shows the performance of G-$\text{NLL}_{SMT}$ compared to G-NLL, which has emerged as the dominant method in Sec. 3.1-3.3. On three out of the four individual tasks, G-$\text{NLL}_{SMT}$ performs better than G-NLL. It also performs better on all of the combinations of tasks, including those that involve unanswerable questions. While the observed gains are only small, they are consistent. In Sec. D.2 we show that the same is true for $\text{MAX}_{SMT}$ and $\text{AVG}_{SMT}$. Furthermore, SMT also improves calibration of G-NLL and MAX on almost all tasks. For example, on *All Combined*, the smoothECE is dropping from 0.147 for G-NLL to 0.091 for G-$\text{NLL}_{SMT}$ and from 0.075 for MAX to 0.053 for $\text{MAX}_{SMT}$ (the impact on AVG is insignificant). The full calibration results can be found in Sec. D.4.

| Task | G-NLL | G-NLL$_{SMT}$ |
|---|---|---|
| Simple | 0.77 ±.045 | 0.78 ±.047 |
| Multiple | 0.76 ±.065 | 0.76 ±.065 |
| Parallel | 0.71 ±.058 | 0.73 ±.057 |
| Parallel-Multiple | 0.76 ±.048 | 0.78 ±.044 |
| Simple + Multiple | 0.77 ±.037 | 0.78 ±.038 |
| Simple + Parallel | 0.75 ±.036 | 0.76 ±.036 |
| Multiple + Parallel-Multiple | 0.77 ±.037 | 0.79 ±.035 |
| All Combined | 0.76 ±.025 | 0.78 ±.025 |
| Simple + Irrelevance | 0.81 ±.025 | 0.82 ±.025 |
| All Combined + Irrelevance | 0.78 ±.020 | 0.80 ±.019 |

Table 3: Mean AUROC, ± mean standard error (mean taken over the eight LLMs) for all tasks from Sec. 3.

Overall, we thus find that one can leverage the FC output particularities to improve over existing UQ methods for LLMs, both for single- and for multi-sample UQ methods.

## 5 Conclusion

We present the first benchmark for UQ methods in LLM Function-Calling (FC) settings. We show that in FC settings, simple logit-based UQ methods largely perform better than more sophisticated multi-sample methods. We further show that the particularities of the FC setting can be utilized to improve existing UQ methods. Specifically, we adapt the semantic-entailment method of multi-sample UQ methods by using Abstract Syntax Trees. For single-sample UQ methods, we use only semantically meaningful tokens, which we define by analyzing FC outputs, when calculating their UQ scores. Both adaptations improve performance over their generic counterparts.

## 6 Limitations

We see two main limitations to this work. First, it would be interesting to compare how our analysis holds for the frontier models that top the BFCL benchmark. However, these tend to not disclose token probabilities through their API, and for those which are open-source, they are computationally infeasible to run on our hardware. Also, we believe that our findings for smaller and medium-sized models are still very relevant, especially in the context of on-device function calling, which would need to rely on such smaller models. Second, it would be interesting to conduct this analysis also for more challenging function-calling settings, such as in multi-turn or agent settings. We leave this analysis for future research.

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

# A    Examples of BFCL Tasks

In this section, we provide illustrative examples of different tasks from the BFCL dataset.

## A.1    Simple

```
{
 "id": "simple_0",
 "question": [
  [
   {
    "role": "user",
    "content": "Find the area of a triangle with a base of 10 units and height of 5 units."
   }
  ]
 ],
 "function": [
  {
   "name": "calculate_triangle_area",
   "description": "Calculate the area of a triangle given its base and height.",
   "parameters": {
    "type": "dict",
    "properties": {
     "base": {
      "type": "integer",
      "description": "The base of the triangle."
     },
     "height": {
      "type": "integer",
      "description": "The height of the triangle."
     },
     "unit": {
      "type": "string",
      "description": "The unit of measure (defaults to 'units' if not specified)"
     }
    },
    "required": [
     "base",
     "height"
    ]
   }
  }
 ]
}
```

## A.2    Multiple

```
{
 "id": "multiple_0",
 "question": [
  [
   {
    "role": "user",
    "content": "Can I find the dimensions and properties of a triangle, if I know its three
        sides are 5 units, 4 units and 3 units long?"
   }
  ]
 ],
 "function": [
  {
   "name": "triangle_properties.get",
   "description": "Retrieve the dimensions, such as area and perimeter, of a triangle if
       lengths of three sides are given.",
```

```
  "parameters": {
   "type": "dict",
   "properties": {
    "side1": {
     "type": "integer",
     "description": "The length of first side of the triangle."
    },
    "side2": {
     "type": "integer",
     "description": "The length of second side of the triangle."
    },
    "side3": {
     "type": "integer",
     "description": "The length of third side of the triangle."
    },
    "get_area": {
     "type": "boolean",
     "description": "A flag to determine whether to calculate the area of triangle. Default
         is true.",
     "default": true,
     "optional": true
    },
    "get_perimeter": {
     "type": "boolean",
     "description": "A flag to determine whether to calculate the perimeter of triangle.
         Default is true.",
     "default": true,
     "optional": true
    },
    "get_angles": {
     "type": "boolean",
     "description": "A flag to determine whether to calculate the internal angles of
         triangle. Default is true.",
     "default": true,
     "optional": true
    }
   },
   "required": [
    "side1",
    "side2",
    "side3"
   ]
  }
 },
 {
  "name": "circle_properties.get",
  "description": "Retrieve the dimensions, such as area and circumference, of a circle if
      radius is given.",
  "parameters": {
   "type": "dict",
   "properties": {
    "radius": {
     "type": "float",
     "description": "The length of radius of the circle."
    },
    "get_area": {
     "type": "boolean",
     "description": "A flag to determine whether to calculate the area of circle. Default
         is true.",
     "default": true,
     "optional": true
    },
    "get_circumference": {
     "type": "boolean",
     "description": "A flag to determine whether to calculate the circumference of circle.
         Default is true.",
     "default": true,
     "optional": true
```

```
    }
   },
   "required": [
    "radius"
   ]
  }
 }
 ]
}
```

## A.3  Parallel

```
{
 "id": "parallel_0",
 "question": [
  [
   {
    "role": "user",
    "content": "Play songs from the artists Taylor Swift and Maroon 5, with a play time of
        20 minutes and 15 minutes respectively, on Spotify."
   }
  ]
 ],
 "function": [
  {
   "name": "spotify.play",
   "description": "Play specific tracks from a given artist for a specific time duration.",
   "parameters": {
    "type": "dict",
    "properties": {
     "artist": {
      "type": "string",
      "description": "The artist whose songs you want to play."
     },
     "duration": {
      "type": "integer",
      "description": "The duration for which the songs should be played, in minutes."
     }
    },
    "required": [
     "artist",
     "duration"
    ]
   }
  }
 ]
}
```

## A.4  Parallel-Multiple

```
{
 "id": "parallel_multiple_0",
 "question": [
  [
   {
    "role": "user",
    "content": "Can you first find out the key historical events related to 'War' and '
        Economy' that took place in France between the years 1800 and 1900? After that,
        could you please tell me the current market value of the sculpture 'The Thinker'
        created by the artist 'Auguste Rodin'? Lastly, I would also like to know the market
        value of the sculpture 'The Kiss', also created by 'Auguste Rodin', in the year
        1882."
   }
```

```
   ]
  ],
  "function": [
   {
    "name": "get_sculpture_value",
    "description": "Retrieve the current market value of a particular sculpture by a specific
         artist.",
    "parameters": {
     "type": "dict",
     "properties": {
      "sculpture": {
       "type": "string",
       "description": "The name of the sculpture."
      },
      "artist": {
       "type": "string",
       "description": "The name of the artist who created the sculpture."
      },
      "required": [
       "sculpture",
       "artist"
      ]
     }
    }
   },
   {
    "name": "history.get_key_events",
    "description": "Retrieve key historical events within a specific period for a certain
         country.",
    "parameters": {
     "type": "dict",
     "properties": {
      "country": {
       "type": "string",
       "description": "The name of the country for which history is queried."
      },
      "start_year": {
       "type": "integer",
       "description": "Start year of the period for which history is queried."
      },
      "end_year": {
       "type": "integer",
       "description": "End year of the period for which history is queried."
      },
      "event_type": {
       "type": "array",
       "items": {
         "type": "string",
         "enum": ["War", "Revolutions", "Diplomacy", "Economy"]
         }
       "description": "Types of event. If none is provided, default that all types will be
           considered."
      },
      "required": [
       "country",
       "start_year",
       "end_year"
      ]
     }
    }
   }
  ]
}
```

## A.5  Irrelevance

```
{
 "id": "irrelevance_0",
 "question": [
  [
   {
    "role": "user",
    "content": "Calculate the area of a triangle given the base is 10 meters and height is 5
        meters."
   }
  ]
 ],
 "function": [
  {
   "name": "determine_body_mass_index",
   "description": "Calculate body mass index given weight and height.",
   "parameters": {
    "type": "dict",
    "properties": {
     "weight": {
      "type": "float",
      "description": "Weight of the individual in kilograms."
     },
     "height": {
      "type": "float",
      "description": "Height of the individual in meters."
     }
    },
    "required": [
     "weight",
     "height"
    ]
   }
  }
 ]
}
```

## B    Example of Natural Questions Answer

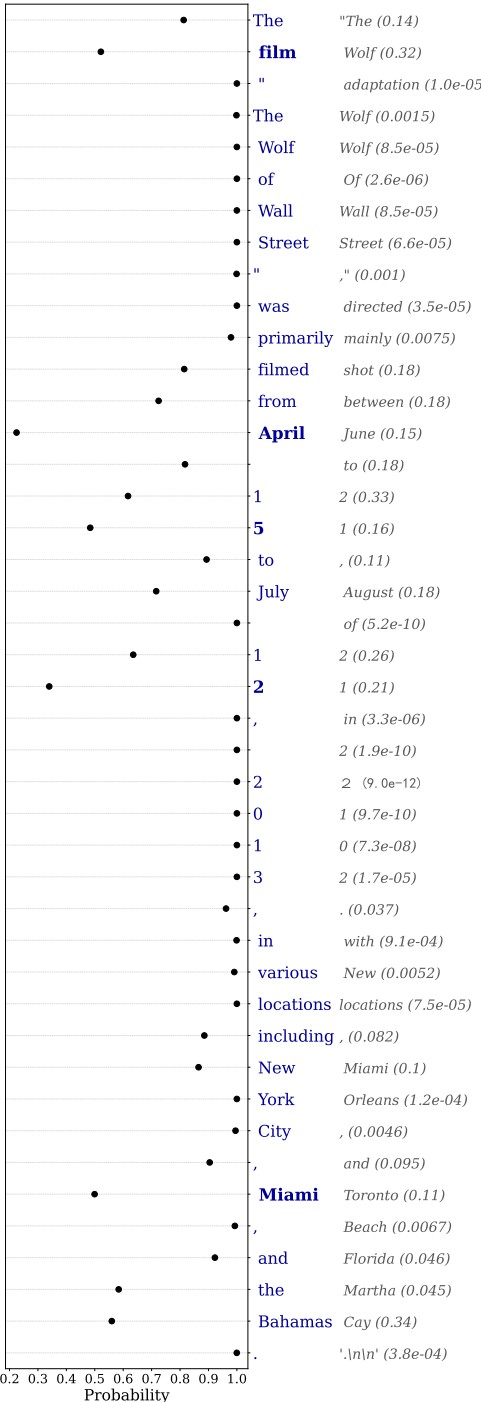

Figure 3: Qwen2.5-7B-Instruct response top token probabilities (lowest 5 bolded) for a question wrongly answered in the Natural Questions dataset.

Question:  *When was the Wolf of Wall Street filmed?*

Ground truth answer:  *The Wolf of Wall Street (2013 film) Filming began on August 8, 2012, in New York. Jonah Hill announced that his first day of shooting was September 4, 2012. Filming also took place in Closter, New Jersey and Harrison, New York. In January 2013, additional scenes were shot at a set built in an abandoned office building in Ardsley, New York. Scenes at the beach house were filmed in Sands Point, New York.*

## C   Uncertainty Estimators

In response to an input question $x$, LLMs output sequences of tokens, so let $y = (y_1, \ldots, a_L)$ denote an output sequence of length $L$. The first family of uncertainty estimation methods uses the greedily decoded sequence, where the LLM always chooses the most likely next token. We denote this sequence with $y^G$. Each token $y_i^G$ comes with a negative log likelihood value (NLL, i.e., token uncertainty) that the LLM has assigned to it during generation based on the previous tokens $y_{<i} = (y_1, \ldots, y_{i-1})$, which we denote as $-\log p(y_i^G | y_{<i}^G, x)$. Single-sample uncertainty methods differ in how they aggregate these probabilities across the sequence into a single uncertainty $u(y^G)$ value for the whole sequence.

**MAX** outputs the highest NLL value in the sequence

$$u_{\text{MAX}}(y^G) := \max - \log p(y_i^G | y_{<i}^G, x). \tag{1}$$

**G-NLL** sums up the NLLs of the sequence (name as per Aichberger et al. (2024)):

$$u_{\text{G-NLL}}(y^G) := \sum_i^L - \log p(y_i^G | y_{<i}^G, x). \tag{2}$$

**AVG** averages the NLLs. It thus normalizes G-NLL by length and can also be interpreted as the logarithm of the sequence perplexity

$$u_{\text{AVG}}(y^G) := -1/L \sum_i^L \log p(y_i^G | y_{<i}^G, x). \tag{3}$$

**LEN** is a baseline that simply outputs the number of tokens in the answer, treating longer sequences as inherently more uncertain. This is an obviously flawed uncertainty estimator and is used more as a sanity-checking baseline that should perform close to random:

$$u_{\text{LEN}}(y^G) := L. \tag{4}$$

The second family of uncertainty estimators samples $J = 10$ responses $(y^1, \ldots, y^J)$ from the LLM, i.e., based on the next-token probabilities with temperature 1. It then combines them into one uncertainty estimate $u((y^1, \ldots, y^J))$ based on different ways of clustering the responses $y^j$.

**PE** does not perform any clustering and treats all outputs as different, even if their decoded strings and even their chosen tokens are the same. This gives a naïve estimate of the predictive entropy of the sequence distribution.

$$u_{\text{PE}}((y^1, \ldots, y^J)) := -H[p(y|x)] \tag{5}$$

$$= E_{y \sim p(y|x)}[\log p(y|x)] \tag{6}$$

$$\approx 1/J \sum_{j=1}^J 1/L_j \sum_i^{L_j} \log p(y_i^j | x) \tag{7}$$

**SE**$_{EXM}$: Semantic Entropy (Farquhar et al., 2024) methods first cluster the output sequences $(y^1, \ldots, y^J)$ and then compute an entropy over the cluster distribution $p(c|x)$. SE$_{EXM}$ uses exact string matching to determine whether two outputs belong to the same cluster, i.e., their decoded text must be exactly equal.

$$u_{\text{SE}}((c^1, \ldots, c^K)) := -H[p(c|x)] \tag{8}$$

$$= E_{c \sim p(c|x)}[\log p(c|x)] \tag{9}$$

$$\approx \sum_{k=1}^K p(c^k|x) \log p(c^k|x), \tag{10}$$

with

$$p(c|x) = \sum_{j=1}^{J} \mathbb{1}\{y^j \in c|x\}p(y^j|x) \ . \tag{11}$$

The cluster probabilities are then normalized to 1, following Kuhn et al. (2023).

**DSE**$_{EXM}$: Discrete Semantic Entropy uses the same clustering algorithm but does not consider the individual sequence likelihoods to compute the cluster distribution $p(c|x)$:

$$p(c|x) = \sum_{j=1}^{J} \mathbb{1}\{y \in c|x\} \ . \tag{12}$$

**SE**$_{AST}$ uses a clustering algorithm that is more suited to the FC setup. In particular, checks whether two decoded function calls are equal up to permutations in their (named) arguments. This is called abstract syntax tree (AST) matching, because it treats two function calls as equivalent if their call will indeed lead to the same inputs to the same function. We first use the AST parser implemented in BFCL to extract the AST of each function call, cast as a dictionary. We then group the calls into clusters based on their key-value pairs' equivalence.

**DSE**$_{AST}$ uses abstract syntax tree matching to form clusters.

**P(True)** Kadavath et al. (2022) look at the probability with which an LLM predicts that $y^G$ is true, given few-shot examples and the set of all generated responses $\{y^1, \ldots, y^J\}$ serving as "brainstormed" alternatives:

$$u_{\text{P(True)}}(y^G) := p(y = \text{A} \mid x_{\text{P(True)}}) \ , \tag{13}$$

where $x_{\text{P(True)}}$ consists of the following parts:

1. System prompt:

```
<|im_start|>system
You are an expert in composing functions. You are given a question and a set of possible
    functions. You are also given brainstormed ideas and a possible answer. Based on
    the question, you have to assess if the possible answer achieves the purpose.

If none of the functions can be used, it should be stated out in the answer. If the
    given question lacks the parameters required by the function, it should also be
    pointed out in the answer. Otherwise, only function calls should be included in the
    answer.

Any invoked function(s) MUST be put it in the format of [func_name1(params_name1=
    params_value1, params_name2=params_value2...), func_name2(params)]
<|im_end|>
```

2. Incorrect few-shot example:

```
<|im_start|>user
Question: [few-shot question inserted here]

Here is a list of functions in JSON format that can be invoked:
[few-shot function inserted here]

Here are some brainstormed ideas:
[few-shot brainstormed ideas inserted here]

Possible answer:
[few-shot incorrect answer inserted here]

Is the possible answer:
A) True
B) False
Respond with A or B only.<|im_end|>
<|im_start|>assistant
The possible answer is: B<|im_end|>
```

3. Correct few-shot example:

```
<|im_start|>user
Question: [few-shot question inserted here]

Here is a list of functions in JSON format that can be invoked:
[few-shot function inserted here]

Here are some brainstormed ideas:
[few-shot brainstormed ideas inserted here]

Possible answer:
[few-shot correct answer inserted here]

Is the possible answer:
A) True
B) False
Respond with A or B only.<|im_end|>
<|im_start|>assistant
The possible answer is: A<|im_end|>
```

4. Actual example for which the correctness of $y^G$ is evaluated:

```
<|im_start|>user
Question: [question inserted here]

Here is a list of functions in JSON format that can be invoked:
[function inserted here]

Here are some brainstormed ideas:
[{y^1,...,y^J} inserted here]

Possible answer:
[y^G inserted here]

Is the possible answer:
A) True
B) False
Respond with A or B only.<|im_end|>
<|im_start|>assistant
The possible answer is:
```

Throughout the p(True) experiments, we use the following few-shot example:

1. Few-shot question:

```
What is 19/53?
```

2. Few-shot function:

```
[{'name': 'divide', 'description': 'Divides two numbers.', 'parameters': {'type': 'dict'
    , 'properties': {'numerator': {'type': 'float', 'description': 'The numerator of the
     fraction.'}, 'denominator': {'type': 'float', 'description': 'The denominator of
    the fraction.'}]}}, 'required': ['numerator', 'denominator']}}, {'name': 'add', '
    description': 'Adds two integers.', 'parameters': {'type': 'dict', 'properties': {'a
    ': {'type': 'int', 'description': 'The first integer.'}, 'b': {'type': 'int', '
    description': 'The second integer.'}}}, 'required': ['a', 'b']}}]
```

3. Few-shot brainstormed ideas:

```
[divide(denominator=53, numerator=19)]
[divide(numerator=53, denominator=53)]
[divide(numerator=19, denominator=19)]
[divide(numerator=19, denominator=53)]
```

4. Few-shot incorrect answer:

```
[divide(numerator=53, denominator=19)]
```

5. Few-shot correct answer:

```
[divide(numerator=19, denominator=53)]
```

# D   Further Experimental Results

## D.1   Entailment-based Semantic Entropy

In Table 4, we show the results of an initial experiment on `Qwen2.5-7B-Instruct` on the *All Combined* task for both the original implementation of SE using DeBERTa as an LLM-based entailment method ($\mathbf{SE}_{ENT}$) to cluster semantically-equivalent generations in comparison to $\mathbf{SE}_{EXM}$ to highlight the inefficacy of the DeBERTa-based approach to FC outputs. Based on this early signal of strongly inferior performance of LLM-based entailment SE in comparison to exact string matching-based SE, and given the additional hours of time required to obtain LLM-based semantic clustering, we opted to go for the exact string matching-based version in the first part of the paper (Sec. 3).

|  | Combined |
|---|---|
| $\mathbf{SE}_{ENT}$ | 0.60 |
| $\mathbf{DSE}_{ENT}$ | 0.60 |
| $\mathbf{SE}_{EXM}$ | 0.75 |
| $\mathbf{DSE}_{EXM}$ | 0.75 |

Table 4: Combined task: AUROC of entailment-based and exact string matching-based SE method for `Qwen/Qwen2.5-7B-Instruct`.

## D.2   Per Model AUROC Results

We present the detailed results of each language model for the individual tasks in Fig. 4, for the combined tasks in Fig. 5, and for the combinations that also include the irrelevance split in Fig. 6.

## D.3   Risk-Coverage Curves

As an additional analysis, we include the risk-coverage curves (averaged across models) for the individual tasks in Fig. 7, for the combined tasks in Fig. 8, and for the combinations that also include the irrelevance split in Fig. 9.

We see that indeed, UQ methods can be used to achieve higher accuracy through abstention. For example, not executing a function call for the 30% of requests with highest G-NLL$_{SMT}$ UQ scores in their output, the accuracies increase from $< 92\%$ to 96% for *Simple*, from 90% to 95% for *Multiple*, from 84% to 90% for *Parallel*, and from 79% to 86% for *Parallel-Multiple* (Fig. 7). The gains are similar for the combinations of tasks, and even reach $\approx +10\%$ once the irrelevance task is added (from 75% to 85% and 80% to 90% in Fig. 9). If one only executes the function calls with the lowest 10% of G-NLL$_{SMT}$ uncertainty scores, accuracies that can be reached are even higher: $\approx 98\%$ for *Simple* and *Multiple*, 95% for *Parallel*, and 93% for *Parallel-Multiple* (Fig. 7), or 97% on *All Combined* (Fig. 8d) and $\approx 95\%$ on the combinations including the irrelevance task (Fig. 9).

Also, we see some differences by method. P$_{true}$ and LEN clearly lag behind, corresponding to their markedly lower AUROCs in Sec. 3. Between the remaining methods, the differences are more nuanced. However, for most tasks, the curves for the single-sample methods (G-NLL and G-NLL$_{SMT}$) are above those of the multi-sample methods (SE$_{EXM}$ and SE$_{AST}$), again corresponding to the AUROC results in Sec. 3. Also, especially for the first half of coverage (until 0.5), the methods adjusted to the FC setting (G-NLL$_{SMT}$ and SE$_{AST}$) have higher curves than those of their generic counterparts. However, this difference typically disappears in the second half of coverage (from 0.5 to 0.1).

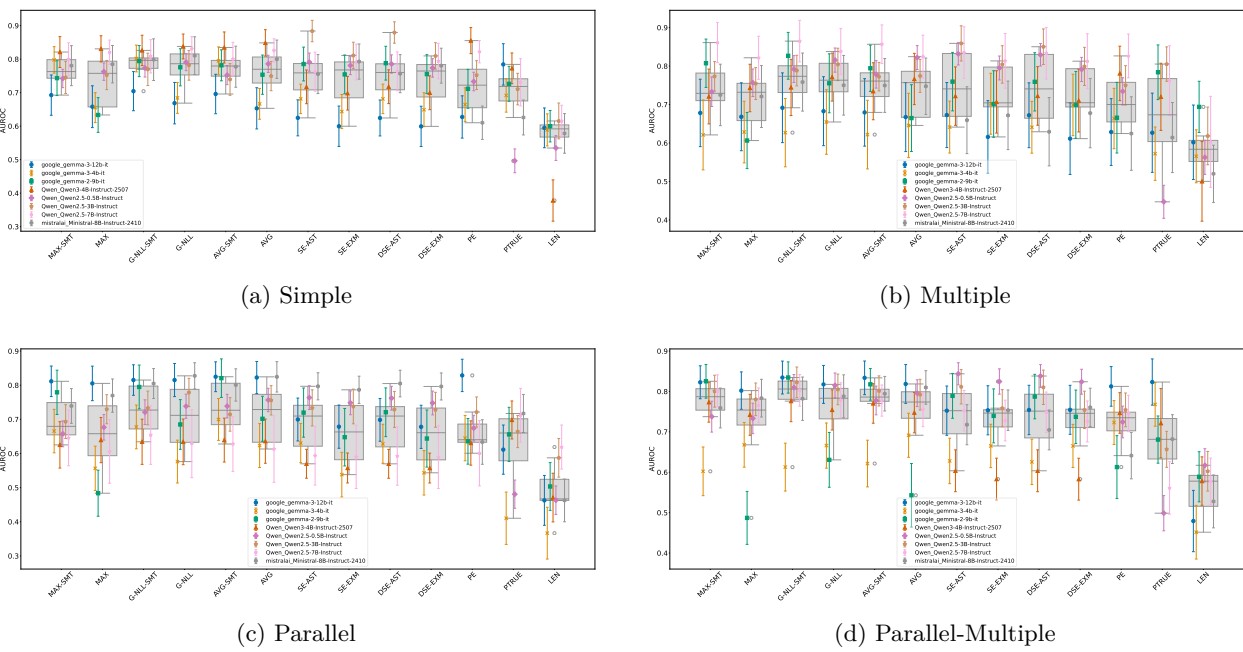

(a) Simple

(b) Multiple

(c) Parallel

(d) Parallel-Multiple

Figure 4: Individual tasks: AUROC $\pm$ standard error (bootstrap estimate with $n = 1000$) for the individual tasks described in Sec. 3.1. Boxplots summarize the variation across LLMs.

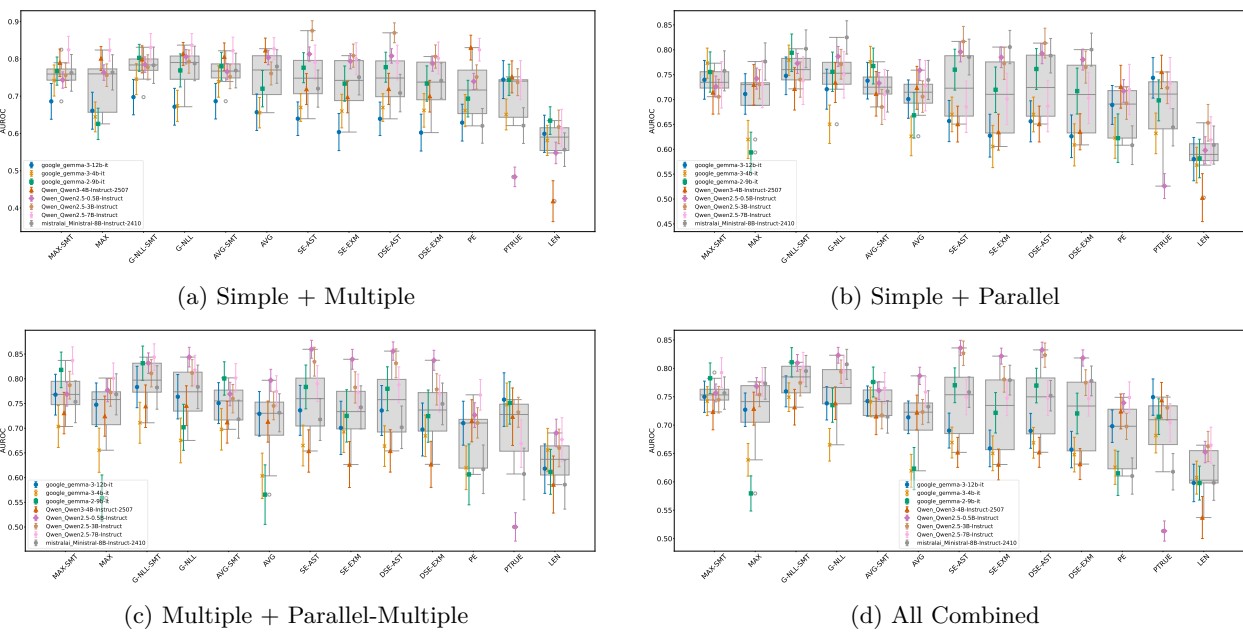

(a) Simple + Multiple

(b) Simple + Parallel

(c) Multiple + Parallel-Multiple

(d) All Combined

Figure 5: Combinations of tasks: AUROC $\pm$ std. error for the combinations of tasks described in Sec. 3.2. Boxplots summarize the variation across LLMs for a given UQ method.

## D.4 Detailed Calibration Results

In Fig. 10 we show the calibration results for all individual and combined task datasets for **MAX**, **MAX**$_{SMT}$, **AVG**, **AVG**$_{SMT}$, **G-NLL**, **G-NLL**$_{SMT}$, and **P**$_{true}$. Other UQ methods do not output probabilities, and their calibration can as such not be evaluated.

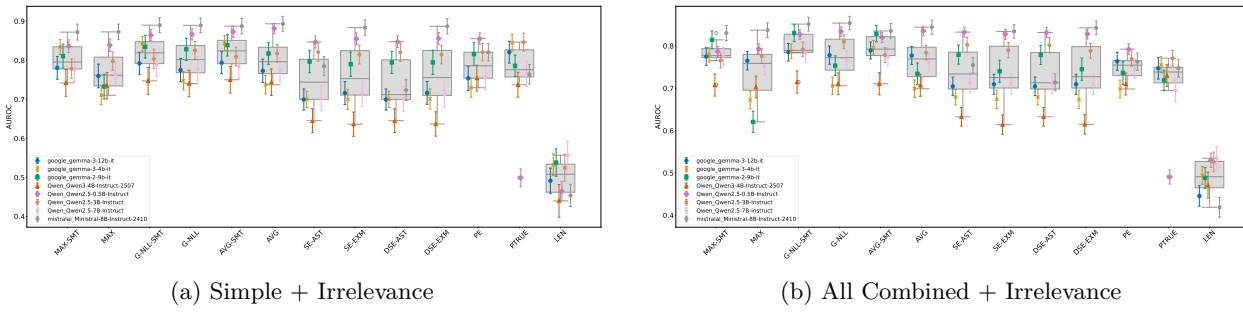

(a) Simple + Irrelevance

(b) All Combined + Irrelevance

Figure 6: Unanswerable requests: AUROC ± standard error for the combinations of answerable and unanswerable tasks described in Sec. 3.3. Boxplots summarize the variation across LLMs for a given UQ method.

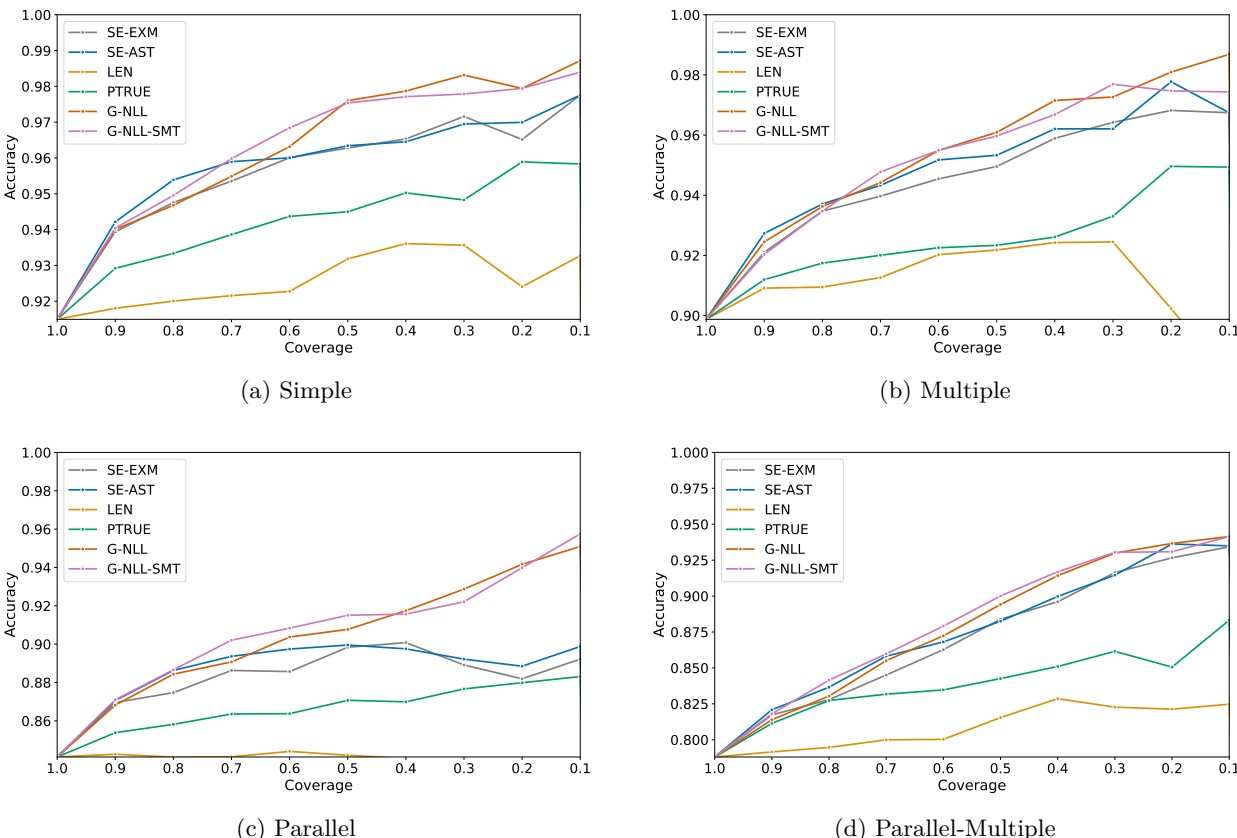

(a) Simple

(b) Multiple

(c) Parallel

(d) Parallel-Multiple

Figure 7: Individual tasks: Risk-coverage curves for the individual tasks described in Sec. 3.1, averaged across models. Higher is better.

# E Compute Infrastructure

## E.1 LLM inference.

LLM inference to generate outputs for the BFCL tasks as described in Sec. 3 was performed using vLLM on a single node with 4 A100 GPUs. Total runtime to generate the greedy samples for 1,240 requests across all task datasets, using 8 models, was roughly 40 minutes in total. Generating 10 high-temperature samples per request required 3 hours in total for all task datasets and models.

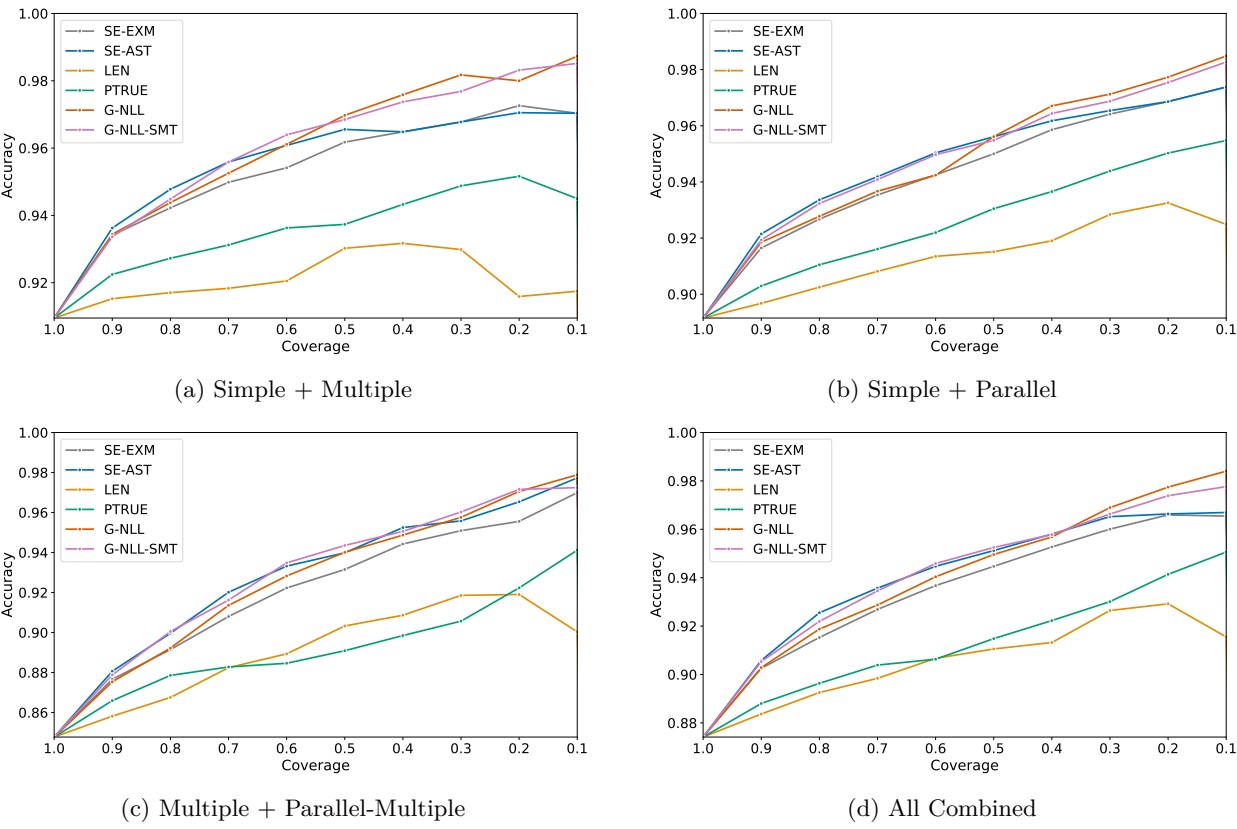

(a) Simple + Multiple      (b) Simple + Parallel

(c) Multiple + Parallel-Multiple      (d) All Combined

Figure 8: Combination of tasks: Risk-coverage curves for the combinations of tasks described in Sec. 3.2, averaged across models. Higher is better.

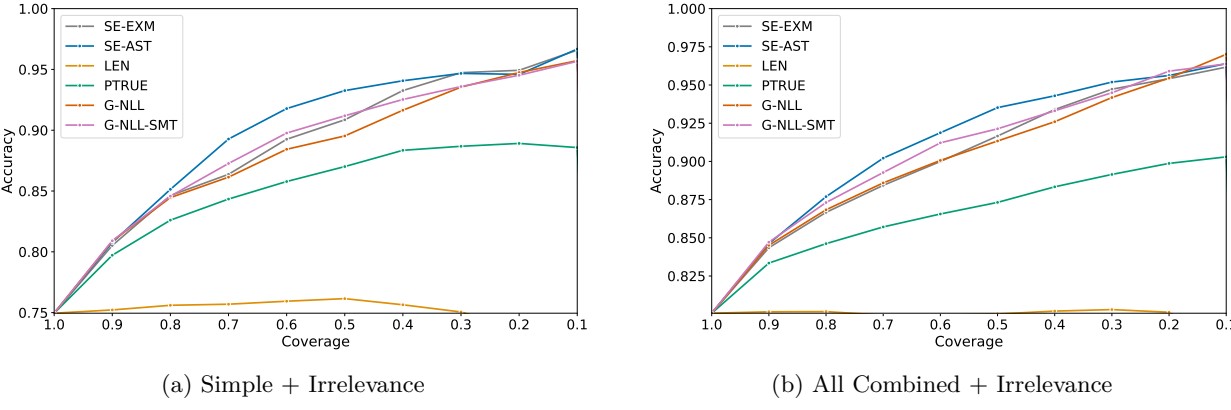

(a) Simple + Irrelevance      (b) All Combined + Irrelevance

Figure 9: Unanswerable requests: Risk-coverage curves for the combinations of answerable and unanswerable tasks described in Sec. 3.3, averaged across models. Higher is better.

## E.2 Computation of UQ scores.

Given the outputs to requests generated by the different LLMs, computation of UQ scores was done locally on a 2021 Apple MacBook Pro with M1 Max chip and 64GB RAM. Computations of all UQ scores for any single model and any (individual or combined) task dataset in our benchmark (see Sec. 3.1 to Sec. 3.3) took less than a minute.

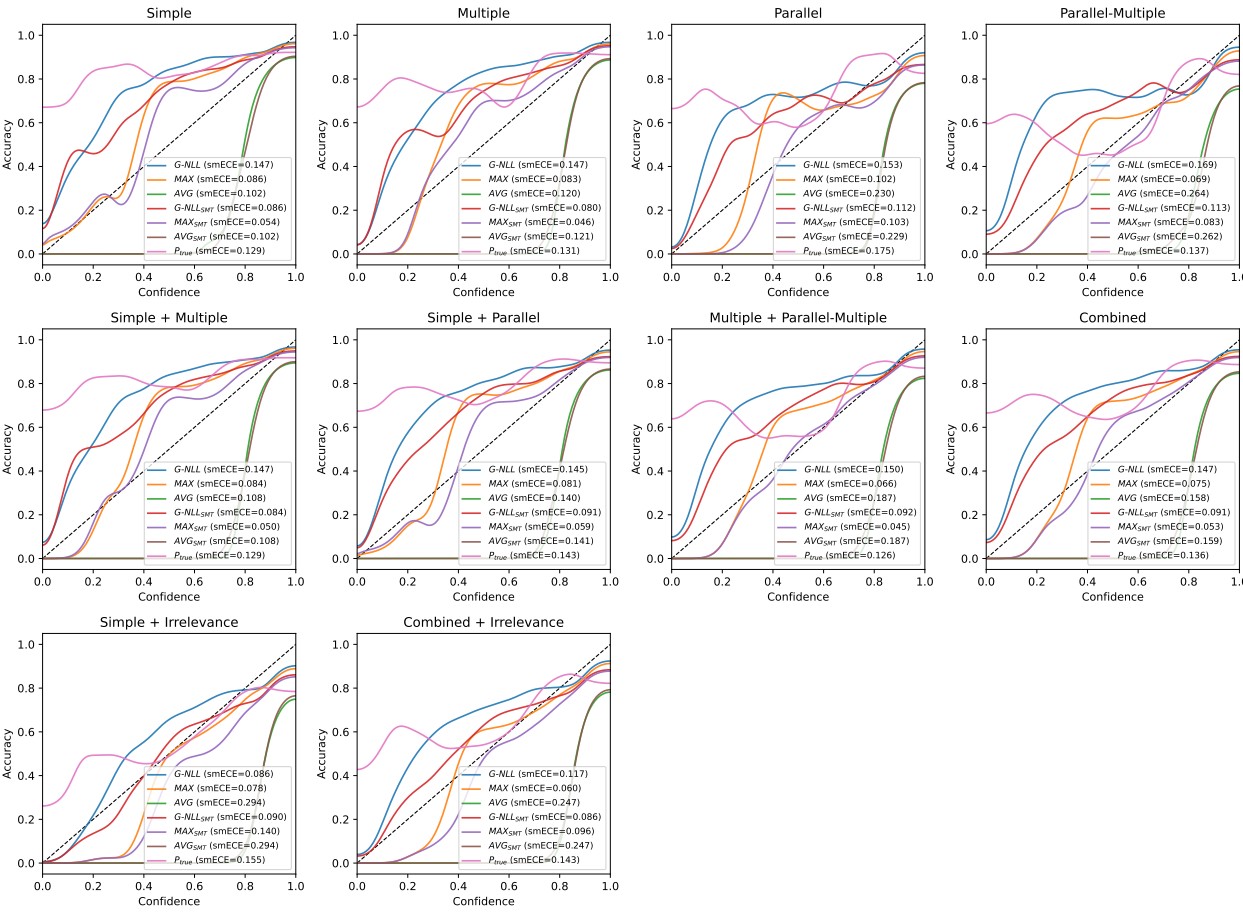

Figure 10: SmoothECE for all tasks aggregated over models.

# F  AST Decoding Errors and Effective Sample Sizes

As described in Sec. 3.1, we exclude requests for which greedy decoding results in an AST decoding error (i.e., the output cannot be interpreted by the Python interpreter). Here, we quantify the frequency of these errors and their impact on our conclusions.

On the *All Combined* split (1000 datapoints in total), 3.4% of the outputs are AST decoding errors, averaged across 8 models. Table 5 shows the effective sample size (number of successfully parsed outputs) per model. We observe that the majority of models produce valid outputs for almost all requests, with the exception of gemma-3-4b-it, which has a notably lower effective sample size of 832.

To assess whether excluding these errors affects our conclusions, we compared the UQ method rankings when including AST decoding errors (marking them as incorrect) versus excluding them. The Spearman rank correlation between the two rankings is 0.94, and the ordering of G-NLL and entropy-based methods stays the same for all. We further note that a non-decodable function call does not result in an actual function execution, and hence it does not pose a risk from the perspective of preventing potentially harmful function calls.

# G  Implementation Details of the Algorithm Extracting Semantically Meaningful Tokens

In Fig. 11 we give the algorithmic implementation of the classification of semantically meaningful tokens as defined in Sec. 4.3 for the python list style function call outputs by the models

Table 5: Effective sample sizes (number of successfully AST-parsed outputs) per model on the *All Combined* split (1000 datapoints total).

| Model | Effective sample size |
|---|---|
| Qwen2.5-0.5B-Instruct | 983 |
| Qwen2.5-3B-Instruct | 980 |
| Qwen2.5-7B-Instruct | 999 |
| Qwen3-4B-Instruct-2507 | 997 |
| Ministral-8B-Instruct-2410 | 987 |
| gemma-2-9b-it | 987 |
| gemma-3-4b-it | 832 |
| gemma-3-12b-it | 962 |

---

**Algorithm 1** SMT typing for Python-call tool outputs

---

**Require:** Token list $T = [t_0, \ldots, t_{N-1}]$, each token has `token_text`
**Ensure:** Same tokens with `token_type` $\in \{\texttt{nfp}, \texttt{nf}, \texttt{np}, \texttt{pv}, \texttt{-}\}$
 1: **for** $i \leftarrow 0$ **to** $N - 1$ **do**
 2:    $t \leftarrow T[i]$
 3:    $t^{-1} \leftarrow T[i-1]$ **if** $i > 0$ **else** None
 4:    $t^{-2} \leftarrow T[i-2]$ **if** $i > 1$ **else** None
 5:    **if** $i = 0$ **or** IsSeparatorOrCloserPython$(t)$ **then**
 6:      $t$.`token_type` $\leftarrow$ `nfp`
 7:    **else if** $i > 0$ **and** IsParamValuePython$(t, t^{-1})$ **then**
 8:      $t$.`token_type` $\leftarrow$ `pv`
 9:    **else if** $i > 0$ **and** IsParamNamePython$(t, t^{-1}, t^{-2})$ **then**
10:      $t$.`token_type` $\leftarrow$ `np`
11:    **else if** IsFuncNamePython$(t, t^{-1}, t^{-2})$ **then**
12:      $t$.`token_type` $\leftarrow$ `nf`
13:    **else**
14:      $t$.`token_type` $\leftarrow$ `-`
15:    **end if**
16: **end for**
17: **return** $T$

---

`qwen/qwen2.5-{0.5,3,7}B-instruct`, `qwen/qwen3-4B-instruct-2507`, `google/gemma-2-9b-it` as well as `google/gemma-3-{4,12}b-it`. In Fig. 12, we do the same for the JSON-style function call outputs by `mistralai/ Ministral-8B-Instruct-2410`.

## H  Sensitivity to Temperature and Number of Samples

In Sec. 4.1, we evaluate multi-sample UQ methods using the default settings of Farquhar et al. (2024): temperature $T = 1.0$ and $J = 10$ samples. Here, we ablate these choices for the best-performing multi-sample method, $\text{SE}_{AST}$.

Table 6 shows the change in mean AUROC (over all models) when varying $T$ while keeping $J = 10$ fixed. Lowering $T$ to 0.5 has a detrimental effect (up to $-0.09$), while increasing to $T = 1.5$ can improve AUROC by up to 0.04. However, increasing $T$ further undoes these improvements and can even lead to a decrease in AUROC compared to $T = 1.0$. For splits involving the *Irrelevance* task, increasing $T$ to 1.5 yields almost no improvement and becomes detrimental at higher temperatures as well.

```python
def classify_token_types(list_of_tokens):

    for idx, token in enumerate(list_of_tokens):
        if idx > 0:
            previous_token = list_of_tokens[idx-1]
        preprevious_token = list_of_tokens[idx-2] if idx > 1 else None

        if (idx == 0) or check_is_num_of_func_or_param(token):
            token['token_type'] = 'nfp'  # Aggregating Categories 1., 5., and 6. in Section 4.2
        elif idx > 0 and check_is_param_value(token, previous_token):
            token['token_type'] = 'pv' # Category 4. in Section 4.2
        elif idx > 0 and check_is_name_of_param(token, previous_token, preprevious_token):
            token['token_type'] = 'np' # Category 3. in Section 4.2
        elif idx > 0 and check_is_name_of_func(token, previous_token, preprevious_token):
            token['token_type'] = 'nf' # Category 2. in Section 4.2
        else:
            token['token_type'] = '-'

    return list_of_tokens
```

(a) The overall algorithm for classifying the tokens in a sequence of tokens.

```python
def check_is_num_of_func_or_param(token):
    if (
        (')' in token['token_text'])
        or (']' in token['token_text'])
        or (',' in token['token_text'])):
        return True
    else:
        return False

def check_is_param_value(token, previous_token):
    if (
        ('=' in previous_token['token_text']) or (previous_token['token_type'] == 'pv')) and (not ((')' in token['token_text']) or (',' in token['token_text']))
        ):
        return True
    else:
        return False

def check_is_name_of_param(token, previous_token, preprevious_token):
    if (
        (previous_token['token_text'].endswith('('))
        or (('(' in token['token_text']) and (not token['token_text'].endswith('(')))
        or ((',' in previous_token['token_text']) and (not (')' in previous_token['token_text'])) and (preprevious_token['token_type'] == 'pv'))
        ):
        return True
    else:
        return False

def check_is_name_of_func(token, previous_token, preprevious_token=None):
    if (
        ('[' in previous_token['token_text'])
        or ((',' in previous_token['token_text']) and (preprevious_token['token_type'] == 'nfp'))
        or (')' in previous_token['token_text'])
        or ((',' in token['token_text']) and (token['token_text'] != ',') and (previous_token['token_type'] == 'nfp'))
        ):
        return True
    else:
        return False
```

(b) The individual token classification functions.

Figure 11: The algorithmic implementation of the classification of semantically meaningful tokens as defined in Sec. 4.3 for the python list style function call outputs by the models `qwen/qwen2.5-{0.5,3,7}B-instruct`, `qwen/qwen3-4B-instruct-2507`, and `google/gemma-2-9b-it` as well as `google/gemma-3-{4,12}b-it`.

Table 7 shows the change in mean AUROC when varying $J$ while keeping $T = 1.0$ fixed. Increasing $J$ to 30 can improve AUROC by around 0.02–0.03, but improvements flatten out beyond that (no better performance for $J = 40$).

Importantly, while the improvements from tuning $T$ and $J$ can close the gap to G-NLL and G-NLL$_{SMT}$, except for the *Multiple* split they do not surpass them. In light of the added computational cost required for multiple samples (Sec. 4.1), our main conclusion remains unchanged: SE as the best multi-sample UQ method offers no clear advantage over G-NLL as the best single-sample method.

```python
def classify_token_types_json(list_of_tokens):

    max_idx = len(list_of_tokens)

    for idx, token in enumerate(list_of_tokens):

        preprevious_token = list_of_tokens[idx-2] if idx > 1 else None
        previous_token = list_of_tokens[idx-1] if idx > 0 else None
        two_token_later = list_of_tokens[idx+2] if idx < (max_idx - 2) else None

        if (idx == 0) or check_is_num_of_func_or_param_json(token, two_token_later):
            token['token_type'] = 'nfp' # Aggregating Categories 1., 5., and 6. in Section 4.2
        elif idx > 1 and check_is_param_value_or_name_of_func_or_param_json(token, preprevious_token):
            token['token_type'] = 'pv' # Aggregating Categories 2., 3., and first tokens of Category 4. in Section 4.2
        elif idx > 0 and fill_param_value_json(token, previous_token):
            token['token_type'] = 'pv' # All remaining tokens of Category 4. in Section 4.2
        else:
            token['token_type'] = '-'

    return list_of_tokens
```

(a) The overall algorithm for classifying the tokens in a sequence of tokens.

```python
def check_is_num_of_func_or_param_json(token, two_token_later):
    if two_token_later:
        if (
            ((']' in token['token_text']) and (token['token_text'] != '[TOOL_CALLS]'))
            or ('}}' in token['token_text']) or ((',' in token['token_text']) and (two_token_later['token_text'] != 'arguments'))
        ):
            return True
    else:
        if ((']' in token['token_text']) and (token['token_text'] != '[TOOL_CALLS]')) or ('}}' in token['token_text']):
            return True
    return False

def check_is_param_value_or_name_of_func_or_param_json(preprevious_token):
    if (
        (':' in preprevious_token['token_text'])
        or ((',' in preprevious_token['token_text']) and (preprevious_token['token_type'] == 'nfp') and
            ('}}' not in preprevious_token['token_text']) and (']}' not in preprevious_token['token_text']))
    ):
        return True
    else:
        return False

def fill_param_value_json(token, previous_token):
    if (not ('"' in token['token_text'])) and (previous_token['token_type'] == 'pv'):
        return True
    else:
        return False
```

(b) The individual token classification functions.

Figure 12: The algorithmic implementation of the classification of semantically meaningful tokens as defined in Sec. 4.3 for the JSON-style function call outputs by mistralai/ Ministral-8B-Instruct-2410.

Table 6: Differences of mean AUROC of SE$_{AST}$ w.r.t. its values for $T\!=\!1.0$. $J\!=\!10$ fixed for all.

| Task | $T\!=\!0.5$ | $T\!=\!1.0$ | $T\!=\!1.5$ | $T\!=\!2.0$ | $T\!=\!3.0$ |
|---|---|---|---|---|---|
| Simple | $-0.09$ | 0 | 0.03 | 0.03 | 0.02 |
| Multiple | $-0.06$ | 0 | 0.03 | 0.02 | 0 |
| Parallel | $-0.01$ | 0 | 0.04 | $-0.01$ | 0.02 |
| Parallel-Multiple | $-0.05$ | 0 | 0.03 | $-0.03$ | $-0.03$ |
| Simple + Multiple | $-0.07$ | 0 | 0.03 | 0.03 | 0.01 |
| Simple + Parallel | $-0.05$ | 0 | 0.04 | 0.01 | 0.01 |
| Multiple + Parallel-Multiple | $-0.06$ | 0 | 0.03 | $-0.01$ | $-0.03$ |
| All Combined | $-0.05$ | 0 | 0.04 | 0 | $-0.01$ |
| Simple + Irrelevance | $-0.07$ | 0 | 0 | 0 | $-0.03$ |
| All Combined + Irrelevance | $-0.06$ | 0 | 0.01 | $-0.01$ | $-0.03$ |

Table 7: Differences of mean AUROC of SE$_{AST}$ w.r.t. its values for $J\!=\!10$. $T\!=\!1.0$ fixed for all.

| Task | $J\!=\!5$ | $J\!=\!10$ | $J\!=\!20$ | $J\!=\!30$ | $J\!=\!40$ |
|---|---|---|---|---|---|
| Simple | $-0.03$ | 0 | 0.01 | 0.02 | 0.02 |
| Multiple | $-0.05$ | 0 | 0.01 | 0.01 | 0.01 |
| Parallel | $-0.04$ | 0 | 0.02 | 0.02 | 0.02 |
| Parallel-Multiple | $-0.03$ | 0 | 0.01 | 0.03 | 0.03 |
| Simple + Multiple | $-0.04$ | 0 | 0.01 | 0.02 | 0.02 |
| Simple + Parallel | $-0.03$ | 0 | 0.02 | 0.02 | 0.02 |
| Multiple + Parallel-Multiple | $-0.04$ | 0 | 0.01 | 0.03 | 0.03 |
| All Combined | $-0.03$ | 0 | 0.02 | 0.03 | 0.03 |
| Simple + Irrelevance | $-0.02$ | 0 | 0.01 | 0.02 | 0.02 |
| All Combined + Irrelevance | $-0.03$ | 0 | 0.01 | 0.02 | 0.02 |

