# OpenReview forum: "Uncertainty Quantification for LLM Function-Calling"
_TMLR — Under review for TMLR_

### Review · Reviewer_LzUu · 2026-06-26

**Summary Of Contributions:**

This paper studies uncertainty quantification for LLM function-calling. The authors build a benchmark based on the Berkeley Function Calling Leaderboard and evaluate several uncertainty estimators, including logit-based single-sample methods, multi-sample entropy-based methods, and Ptrue. The main empirical finding is that simple single-sample logit-based methods, especially G-NLL, are competitive with or better than multi-sample methods such as Semantic Entropy in this function-calling setting. The paper further proposes two function-calling-specific adaptations: clustering sampled outputs using AST matching for multi-sample methods, and computing single-sample uncertainty only over semantically meaningful tokens.

The problem is important and timely, since function-calling errors can have much more direct consequences than ordinary text-generation errors. The paper is also clearly written, and the qualitative analysis of token probabilities in function-calling outputs is useful. However, I have several concerns about the current evidence. The evaluation is relatively narrow compared to the broad framing of uncertainty quantification for LLM function-calling, especially because it focuses heavily on white-box logit-based methods and BFCL-style single-turn tasks. I also find the comparison around Semantic Entropy somewhat incomplete, and many reported differences between methods are small, making it hard to assess the practical significance of the findings.

**Additional Comments:**

The paper is clearly written and the qualitative analysis in Section 4.2 is helpful. I especially liked the observation that function-calling outputs have a different token-probability structure from natural language answers. My main concern is not the motivation, but the breadth and strength of the empirical evidence. I think the paper would become substantially stronger with a broader set of black-box baselines, more deployment-oriented metrics, and a clearer assessment of whether the small differences between methods are practically meaningful.

**Audience:**

Yes

**Audience Explanation:**

The topic is relevant to a significant part of the TMLR audience. Function-calling and tool-use are increasingly important for LLM deployment, and uncertainty estimation for such outputs is an important reliability problem. The finding that simple token-likelihood scores can be competitive with more complex multi-sample methods in structured function-calling outputs is potentially useful, although I think it could be trivial. The qualitative observation that function-calling outputs often have very high token probabilities except for a few semantically important positions is also interesting.

That said, the current version would be more useful if it better separated the robust findings from benchmark-specific observations. In particular, I think the paper should be more cautious about broad conclusions until it includes stronger black-box baselines, more deployment-relevant metrics, and a clearer analysis of the small differences between methods.

**Broader Impact Concerns:**

I do not have major broader impact concerns beyond the standard concerns for LLM function-calling and tool-use systems. The paper is motivated by reducing harmful or incorrect tool executions, which is a positive direction. However, because the work is about deciding when to execute or abstain from function calls, the authors should be careful to discuss failure modes where uncertainty estimates are overconfident.

**Claims And Evidence:**

No

**Claims Explanation:**

The paper provides evidence for a narrower claim: on the authors’ BFCL-derived benchmark, simple logit-based uncertainty scores are competitive with the tested multi-sample entropy methods, and function-calling-specific normalization or token selection can give small improvements. I find this part mostly supported.

However, I do not think the current evidence fully supports the broader claims about uncertainty quantification for LLM function-calling. First, the evaluation is dominated by white-box methods that require token probabilities. This is a serious limitation for function-calling, where many practical systems expose only sampled outputs or final tool calls. The paper includes Ptrue, but this is not enough to cover the space of black-box uncertainty methods. Missing baselines include self-consistency, sample agreement, AST-level agreement among sampled tool calls, model-verbalized confidence, and possibly judge-based or embedding-based confidence methods.

Second, the treatment of Semantic Entropy is not fully convincing. Exact-match clustering removes much of the semantic component of Semantic Entropy and makes the method closer to entropy over repeated surface forms. The AST-based variant is useful, but the paper remains too centered on Semantic Entropy without comparing against stronger semantic or structural clustering alternatives. If LLM-based entailment performs poorly for function calls, this should be reported more directly as a negative result. Embedding-based or judge-based clustering may also be feasible, especially given the small number of samples used.

Third, many results are very close across methods. The proposed adaptations improve performance, but the gains are often small. The paper would benefit from stronger statistical analysis, clearer effect sizes, and more discussion of when the methods meaningfully differ. As written, it is hard to tell whether the main conclusion is that G-NLL is genuinely preferable, or that most uncertainty scores behave similarly on this benchmark.

Finally, the evaluation relies too heavily on AUROC. AUROC is a useful ranking metric, but in function-calling the deployment question is often about abstention under asymmetric costs: how many incorrect calls can be filtered at a fixed coverage, or how much coverage is possible under a target error rate. Metrics such as AUPRC, AURC, risk at fixed coverage, coverage at fixed risk, and cost-sensitive analyses would make the evidence much more convincing.

**Requested Changes:**

Here are some suggestions:

1. Please add some stronger black-box uncertainty baselines. The current evaluation is too dominated by white-box logit-based methods. This is a significant limitation because many practical function-calling systems do not expose token probabilities. The paper should include baselines such as self-consistency.
2. Strengthen the comparison around Semantic Entropy. I am not convinced that exact-match clustering is an adequate representation of Semantic Entropy, since it removes most of the semantic clustering component. The AST-based variant is useful, but the paper should more clearly distinguish exact-match entropy from semantic entropy and compare against stronger clustering alternatives, such as embedding-based clustering, LLM-judge clustering, or normalized AST-based self-consistency. If entailment-based clustering performs poorly for function-calling, this should be reported clearly as an important negative result.
3. It’s good to add deployment-relevant evaluation metrics beyond AUROC. AUROC is useful, but it is not sufficient for evaluating uncertainty in function-calling systems. The paper should report metrics such as AUPRC.
4. The authors should clarify more on the practical significance of the reported differences. Many methods achieve similar AUROC values, and the proposed function-calling-specific adaptations often give small gains. The paper should include paired significance tests, clearer effect sizes, and more analysis of where the methods disagree. This would make it easier to determine whether the differences are robust and practically meaningful.
5. Narrow or qualify the main claims. The current evidence supports a more modest claim: simple logit-based uncertainty scores work well on this BFCL-derived single-turn function-calling setup, and function-calling-specific adaptations can slightly improve them. Broader claims about uncertainty quantification for LLM function-calling should be qualified unless the evaluation is expanded.

Here are some additional things that would strengthen the work:

1. Include more diverse function-calling settings, especially multi-turn or agentic tasks.
2. Report more detailed per-model and per-task failure cases, especially cases where multi-sample methods and single-sample methods disagree.
3. Clarify the computational cost of the proposed methods, including the cost of sampling, clustering, and semantically meaningful token extraction.
4. Discuss how thresholds would be chosen in practice for abstention before executing function calls.
5. Provide more analysis of whether the semantically meaningful token heuristic is robust across different function-call formats.

---

### Review · Reviewer_inB4 · 2026-06-28

**Summary Of Contributions:**

The paper studies uncertainty quantification (UQ) for the LLM Function-Calling (FC) / tool-use setting, motivated by the safety argument that incorrect tool calls can trigger irreversible actions (transferring money, deleting data), so a system should be able to estimate its confidence before executing a call. The authors make three main contributions:

1. **A UQ benchmark for FC built on BFCL.** The authors repurpose the Berkeley Function Calling Leaderboard (BFCL) for a *selective-prediction* evaluation: correctness is determined by AST-matching against the ground-truth call, the problem is cast as binary classification of correct vs. incorrect, and a UQ score is used as the classifier. They evaluate a set of UQ baselines across eight open instruction-tuned models (0.5B–12B): single-sample methods (maximum token negative log-likelihood, average token negative log-likelihood, sequence negative log-likelihood (G-NLL), and a sequence-length sanity baseline); multi-sample methods (predictive entropy, semantic entropy (SE), and discrete semantic entropy (DSE) with exact-string-matching clusters); and the prompting-based P(True).

2. **Central empirical findings:**

   - Because FC outputs are highly constrained, many token probabilities are very high, so sampled generations collapse into a single semantic cluster, which deflates entropy-based scores even for wrong calls.
   - Not all tokens in a generation contribute meaningfully to uncertainty; only the semantically meaningful ones do, which the single-sample adaptation exploits.

3. **Two FC-tailored adaptations of existing methods:**

   - For multi-sample methods, AST-based clustering replaces exact string matching, improving the basic SE/DSE scores on 8/10 tasks.
   - For single-sample methods, a "Semantically Meaningful Tokens" (SMT) scheme aggregates log-probs over only the tokens that carry decision-relevant semantics (extracted by a rule-based, format-specific parser), yielding small AUROC gains and improved calibration.

**Key strengths.** (a) A genuinely under-explored and timely problem with a clear safety motivation. (b) Interesting experimental results illustrating the limitations of existing UQ methods in the FC setting. (c) Useful suggestions for improving UQ for FC in the agentic setup. (d) Sensible ablation studies and checks (bootstrap SEs, AST-error ablation with Spearman 0.94, temperature/sample-count ablations, calibration). (e) Several interesting setups in the developed benchmark.

**Key weaknesses.** (a) Semantic Entropy is a well-established but relatively old method, and prior work has shown that it can be outperformed by simpler likelihood-based scores such as negative log-likelihood, perplexity, or token entropy, as well as by lexical-similarity methods (e.g., Vashurin et al., 2025). In some cases, simple methods such as Lexical Similarity (Fomicheva et al., 2020) also outperform SE. As a result, the paper's premise that Semantic Entropy is the strongest multi-sample baseline remains insufficiently tested against similarity-graph and representation-based estimators that are often competitive with, or superior to, it. I would specifically suggest evaluating CoCoA, which combines AST-based matching with sequence log-likelihood. (b) AUROC may not be the most informative metric for selective prediction: prediction-rejection curves better show whether the reported improvements are practically meaningful, since some gains may arise only at unrealistically high rejection rates. Specifically, in Figure 8, SE + AST actually outperforms G-NLL in the first half of the rejection curve. Moreover, PR-AUC can be misleading under strong class imbalance, as appears to be the case here. Reporting PRR at fixed rejection rates (e.g., 30% or 50%) would make the evaluation more credible and easier to interpret, and would likely change some of the claims regarding sampling-based methods. (c) Some of the improvements supporting Contribution 3 fall within the reported standard errors and are not accompanied by paired significance tests; the claim that the proposed adaptations "improve performance" is therefore currently suggestive rather than firmly established.

**References**

Fomicheva, Marina, Shuo Sun, Lisa Yankovskaya, Frédéric Blain, Francisco Guzmán, Mark Fishel, Nikolaos Aletras, Vishrav Chaudhary, and Lucia Specia. 2020. *Unsupervised Quality Estimation for Neural Machine Translation*. Transactions of the Association for Computational Linguistics.

Lin, Zhen, Shubhendu Trivedi, and Jimeng Sun. 2023. *Generating with Confidence: Uncertainty Quantification for Black-box Large Language Models*. arXiv:2305.19187.

Vashurin, Roman, Ekaterina Fadeeva, Artem Vazhentsev, Lyudmila Rvanova, Daniil Vasilev, Akim Tsvigun, Sergey Petrakov, Rui Xing, Abdelrahman Sadallah, Kirill Grishchenkov, Alexander Panchenko, Timothy Baldwin, Preslav Nakov, Maxim Panov, and Artem Shelmanov. 2025. *Benchmarking Uncertainty Quantification Methods for Large Language Models with LM-Polygraph*. Transactions of the Association for Computational Linguistics.

**Additional Comments:**

n/a

**Audience:**

Yes

**Audience Explanation:**

UQ for tool-use sits at the intersection of two areas of active TMLR interest: uncertainty quantification for generative models and LLM-agent reliability.

**Claims And Evidence:**

Yes

**Claims Explanation:**

**Partially.** The claim that sampling-based methods are inferior to NLL or other single-sample techniques is too broad as currently stated. CoCoA, which combines sampling-based similarity (e.g., AST matching) with sequence probability, should not be worse, and it is closer to a credibility-estimation technique than to a pure uncertainty measure. The deeper issue is that SE has previously been shown to be inferior to basic techniques in many settings (see Vashurin et al., 2025): SE is an uncertainty quantification method, whereas a method that estimates the credibility of a particular generation would be more appropriate here. Separately, the claim that the adaptations improve performance (Contribution 3) would be stronger with statistical testing.

**Requested Changes:**

### Critical

1. Add experiments with other sampling-based techniques — DegMat, Eccentricity, or EigenValue (Lin et al., 2023) — using AST-equivalence as the similarity for FC. I also strongly recommend adding CoCoA (AST-matching similarity plus sequence probability) as a baseline. It would further be useful to test sampling-based methods that use latent representations: since output-space signals collapse in FC, representation-based estimators (e.g., EigenScore/INSIDE over hidden-state covariance, or latent density scores) may help; they reuse the existing sample budget, and a small pilot would complement the SMT analysis. Accordingly, the claims regarding the performance of sampling-based techniques should be softened.

2. Report the statistical significance of the improvement claims. The SE_AST-over-SE_EXM and SMT-over-baseline gains fall within the reported standard errors; it would be useful to add a paired test for G-NLL_SMT vs. G-NLL, SE_AST vs. SE_EXM, and best single- vs. best multi-sample.

3. Please compute PRR@30 or PRR@50; as Figure 8 shows, sampling-based methods are actually better than G-NLL in the first part of the curve.

### Would strengthen

1. Add graphical illustrations for the different regimes (Simple, Parallel, Parallel-Multiple, etc.) to improve clarity.
2. Code/benchmark release. Please state clearly whether the authors plan to release the benchmark harness, the UQ implementations, the generated model outputs / UQ scores, and the SMT extractors. Since the paper's first contribution is a benchmark for UQ in function-calling, reproducibility depends on the availability of these artifacts.
3. Add one sentence noting that SMT extraction is label-free (it uses only the output's structure, not the gold call), to preempt label-leakage concerns.
4. AST clustering does not help (and slightly hurts) on the Irrelevance splits, and the FC-adapted advantage vanishes in the second half of coverage; a brief discussion would improve completeness.
5. Page 6: "Table 2 suggest that" → "Table 2 suggests that".

---

### Review · Reviewer_eW8n · 2026-06-29

**Summary Of Contributions:**

This paper presents a systematic evaluation of uncertainty quantification (UQ) methods for LLM function calling. The evaluation is conducted on the BFCL tasks, where the task consists of selecting the correct function(s) and arguments from a set of function definitions provided in context. While this setup is limited compared to modern agentic systems, it provides a clean and well-defined benchmark for studying uncertainty estimation.

The paper evaluates a broad set of existing UQ methods that fall into two categories - (i)  single-sample methods, which estimate uncertainty from a single generated function call using token-level probabilities (requiring models that expose token probabilities) and (ii) consists of multi-sample methods that generate multiple outputs and estimate uncertainty using predictive or semantic entropy after clustering semantically equivalent outputs.

The main empirical finding is that, unlike previous observations in NLG, multi-sample methods do not outperform simple logit-based single-sample methods for function calling. In particular, G-NLL consistently performs as well as or better than Semantic Entropy across the evaluated settings. The paper further evaluates calibration and selective prediction (failure detection through abstention) and shows that G-NLL provides competitive performance without requiring expensive multi-sample inference.

The authors then investigate why multi-sample methods underperform in this setting and propose an AST-based clustering strategy that better accounts for semantic equivalence between function calls. Since single-sample methods already perform best, they additionally propose computing uncertainty only over "Semantically Meaningful Tokens" (SMTs), extracted using a rule-based parser for function-call outputs. This modification yields small improvements over standard G-NLL.

**Strengths**
- The paper is exceptionally well written and easy to follow, with a logical progression from empirical observations to hypotheses and proposed improvements.
- The paper provides useful empirical insight into why uncertainty estimation behaves differently for structured function-calling outputs than for natural language generation.

**Weaknesses**

- The evaluation is restricted to a relatively narrow form of function calling where all candidate functions are provided in context. This setup is considerably simpler than modern tool-use settings involving MCP servers, retrieval, planning, or multi-step agents, limiting the generality of the conclusions.
- The discussion of failure detection would benefit from a stronger presentation of addl literature . There is a substantial work showing that raw uncertainty scores alone are often insufficient for reliable failure detection, and the paper does not investigate post-hoc calibration methods or calibrated decision thresholds.
- Next, the proposed SMT method yields only modest performance improvements. They are numerically small, and the paper does not provide statistical significance (e.g., p-values) to establish whether the observed improvements are meaningful across models and tasks.
- Lastly, there is no treatment of the classical UQ literature in the decomposition of UQ into aleatoric and epistemic components

**Audience:**

Yes

**Audience Explanation:**

Yes. I believe the findings would be of interest to researchers working on uncertainty quantification, LLM reliability, and tool-using language models.  Even though the methodological contributions are incremental, the benchmark and empirical analysis are likely to be useful to the community.

**Broader Impact Concerns:**

None.

**Claims And Evidence:**

Yes

**Claims Explanation:**

The main claims are mostly supported by accurate and clear evidence. The authors provide a systematic evaluation across multiple models, BFCL task variants including irrelevance, single-sample and multi-sample UQ methods and , calibration analyses.The evidence convincingly supports the central empirical claim that, in this benchmark, simple token-probability-based methods such as G-NLL are competitive with or better than multi-sample semantic entropy methods.

The evidence for the proposed adaptations is also clear, but less convincing in terms of practical impact. AST-based clustering improves multi-sample methods in several settings, and the SMT-based variant improves G-NLL consistently. However, the gains are quite small. I would like to see stronger statistical analysis, to establish whether these improvements are significant and robust.
Similarly, the discussion of failure detection would be more convincing if the authors studied post-hoc calibration, since raw UQ scores are not usually sufficient for reliable deployment.

Overall, the empirical evidence is clear and supports the paper’s main conclusions within the evaluated setting, but some claims about practical failure detection and general function-calling reliability should be more carefully scoped including the title.

**Requested Changes:**

1. The SMT-based uncertainty estimator produces small but higher AUROC values than vanilla G-NLL.  The paper reports mean AUROC together with averaged standard errors, but this does not seem to establish whether the improvements are statistically significant. Since SMT is presented as a methodological contribution, I encourage the authors to perform a paired statistical analysis (e.g., paired significance tests comparing G-NLL and G-NLL-SMT) or otherwise moderate the claims regarding its empirical advantage.
2.  The paper motivates uncertainty estimation as a mechanism for failure detection and deciding whether to execute function calls. While AUROC appropriately measures ranking quality, practical deployment generally requires calibrated confidence estimates and decision thresholds. I encourage the authors to discuss or evaluate a simple post-hoc calibration method (e.g.,  temperature scaling) and revisit the utility of the UQ methods under a post-hoc calibration. The findings could change and it would be important to realize that.
3. The paper already acknowledges that the evaluation is restricted to BFCL-style function calling and leaves more complex agentic settings to future work. It would nevertheless be useful to briefly discuss which aspects of the findings are expected to transfer to richer tool-use settings and which may depend on the constrained benchmark.